computational biology

phylogenetic signal, evolutionary rate, structural features, ancestral state reconstruction, Transeurasian, Altaic

**Author for correspondence:**
Nataliia Hübler
e-mail: nataliia_huebler@eva.mpg.de

# Phylogenetic signal and rate of evolutionary change in language structures

Nataliia Hübler[1,2]

[1]Department of Linguistic and Cultural Evolution, Max Planck Institute for the Science of Human History, Kahlaische Str. 10, Jena 07745, Germany
[2]Department of Linguistic and Cultural Evolution, Max Planck Institute for Evolutionary Anthropology, Deutscher Platz 6, Leipzig 04103

NH, 0000-0002-0013-563X

Within linguistics, there is an ongoing debate about whether some language structures remain stable over time, which structures these are and whether they can be used to uncover the relationships between languages. However, there is no consensus on the definition of the term 'stability'. I define 'stability' as a high phylogenetic signal and a low rate of change. I use metric $D$ to measure the phylogenetic signal and Hidden Markov Model to calculate the evolutionary rate for 171 structural features coded for 12 Japonic, 2 Koreanic, 14 Mongolic, 11 Tungusic and 21 Turkic languages. To more deeply investigate the differences in evolutionary dynamics of structural features across areas of grammar, I divide the features into 4 language domains, 13 functional categories and 9 parts of speech. My results suggest that there is a correlation between the phylogenetic signal and evolutionary rate and that, overall, two-thirds of the features have a high phylogenetic signal and over a half of the features evolve at a slow rate. Specifically, argument marking (flagging and indexing), derivation and valency appear to be the most stable functional categories, pronouns and nouns the most stable parts of speech, and phonological and morphological levels the most stable language domains.

## 1. Introduction

Tracing the history of languages and their speakers is a challenging undertaking. Linguists draw on various aspects of language to track these histories, often relying on 'basic' or 'core' vocabulary as a marker of language history [1–5]. However, recently more studies have been using structural features of languages to answer questions about language history and population movements [6–12].

Although it seems clear that structural features provide another source of information on the history of languages, there is an ongoing debate about whether they can recover history as well as or at a deeper level than basic vocabulary [6, p. 2073]; [11, pp. 1–2].

Some critiques argue that language structures primarily reflect the history of contact between the languages in question due to their susceptibility to borrowing and high rates of chance similarities given a limited set of states, often only 'present' or 'absent' [13, p. 3923] [8]. In fact, the stability of a set of structural features cannot be directly compared with the stability of a basic vocabulary list, as the latter was preselected for the known stability of the concepts [14, p. 122]; [15, p. 68]. A set of structural features can rather be compared with a random list of words in a language, where basic concepts and words with a high borrowability level are included.

To tackle the problem described above, there have been several attempts at defining a set of stable structural features. Nichols [16, pp. 209–210] points to items she claims are stable, comprising inclusive/exclusive opposition, head/dependent marking and alignment. Greenhill et al. [12] compared the rate of change in basic vocabulary and structural features and came to a conclusion that structural features (grammar and phonology) change faster than basic vocabulary on average. Nevertheless, they state that there is a core of grammatical features that evolve at a slow rate. Dediu & Levinson [11] constructed stability profiles for the features from World Atlas of Language Structures [17]; however, Greenhill et al. [18, p. 2449] argue that these data have serious limitations due to coding scheme (high level of categorization in WALS versus direct presence/absence coding in the current study, which follows the guidelines of Grambank [19]).

A major problem is that stability is a complex concept, often conflated with either phylogenetic signal (i.e. how well a given trait fits onto a given language phylogeny) or with evolutionary rate (i.e. how fast a trait changes on a given phylogeny). Because of this complexity, Revell et al. [20, p. 591] strongly encourage studies to treat stability (or conservatism), phylogenetic signal and evolutionary rate separately, as their results suggest that there is no correlation between either of these. Instead, Revell et al. [20, p. 591] define, for evolutionary biology, phylogenetic signal as 'the statistical non-independence among species trait values due to their phylogenetic relatedness'. For our purposes, we could translate this definition in linguistics terms: if a feature value of one language depends on the feature value of another language due to the relatedness of these languages, then the feature has a phylogenetic signal.

As for evolutionary rate, I investigate both directions of change, feature loss and feature gain, and calculate not only the transition rates between the two states, present and absent, but also the probability of states being absent or present at the nodes corresponding to proto-languages. Reconstructing some parts of the vocabulary and the phonological system of a proto-language is an ordinary procedure in comparative linguistics, but the field of linguistic and cultural evolution is still far from routinely reconstructing grammatical structures or cultural phenomena to ancestral stages. So far, there are only several studies using phylogenetic comparative methods to reconstruct individual abstract aspects of language, comprising word order [21], numeral systems [22], colour terms [23] and Indo-European grammar [24].

The two main competing forces in language evolution are inheritance and language contact, therefore it is not sufficient for our understanding of the evolutionary dynamics of structural features to study language families with a highly tree-like structure [13] and a low conflicting signal to conclude that some structural features evolve at a slow rate [24, p. 586]—we need to compare the performance of structural features across different language families, with high and low proportions of borrowing in their languages. The languages spoken across (mainly) Northern Asia provide a perfect sample for this endeavour, because they cover a large enough area, with well-known contact relations between them, and exhibit the perfect degree of genealogical heterogeneity. The languages are known to share a set of typological similarities, but the source of these similarities is unclear: there are hypotheses suggesting their genealogical relationship [25,26] as well as studies discarding these hypotheses and attributing the similarities to borrowing and chance [27,28] (the lists of the supporters and critics are by no means exhaustive). Even though the approach in this study does not aim at proving or discarding any hypotheses on the relatedness of these language families, the language sample is nevertheless highly suitable for investigating stability of structural features because of the known areal effects. If, despite the high levels of contact between the languages in the sample and a high potential for feature transfer, we can show that some structural features have a phylogenetic signal, then it would indicate that structural features convey a historical signal that is due to genealogical relationships rather than language contact.

# 2. Material and methods

## 2.1. Materials

The language sample contains languages belonging to five language families: 12 Japonic, 2 Koreanic, 14 Mongolic, 11 Tungusic, 21 Turkic languages (see figure 1 for the geographical distribution of the

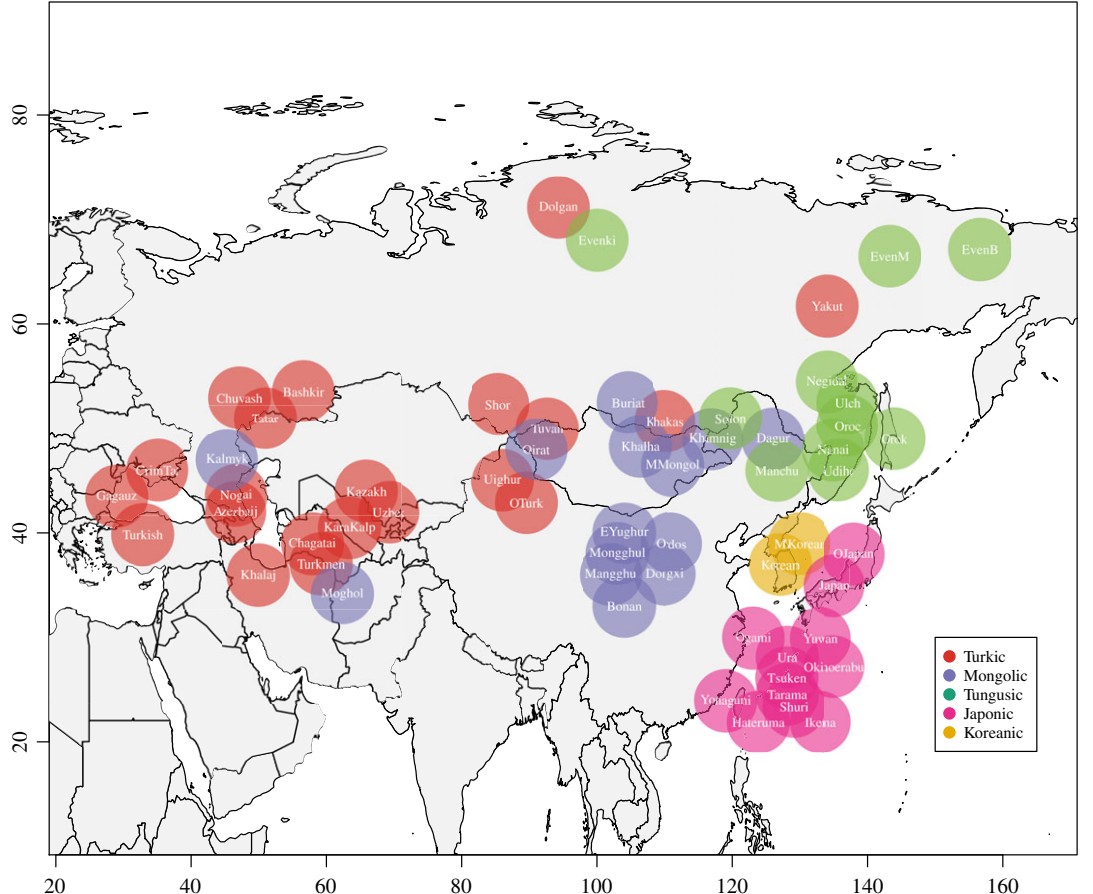

**Figure 1.** Distribution of the languages considered in the study. Some language names are represented as short versions of full names: Azerbaij = Azerbaijani, CrimTat = Crimean Tatar, Dongxi = Dongxiang, EvenB = Beryozovka Even, EvenM = Moma Even, KaraKalp = Kara-Kalpak, Khamnig = Khamnigan, MKorean = Middle Korean, MMongol = Middle Mongol, OJapan = Eastern Old Japanese, OTurk = Old Turkic, Uzbek = Northern Uzbek. Coordinates of languages adapted from Glottolog [29].

languages). Each language[1] was coded for 224 features. I used 189 structural features from the Grambank database [19] and binarized six Grambank features on word order (from 'What is the order of X and Y?' to 'Can X precede Y?' and 'Can Y precede X?'). I added 35 features on phonology and formal representation to increase the variability among language families. Eight of these feature formulations (TE004–TE008, TE018, TE019, TE027) are based on the feature set from Robbeets [30] and show some variation in the region. Each feature received the value 1, if the feature question could be answered with a 'yes', and the value 0, if the feature question could be answered with a 'no'. If there was not enough information on the feature in the grammar, the feature was coded as '?' (replaced by 'NA' in further analysis). Out of 224 features, 53 features were absent in all languages and were therefore excluded from further analysis (the algorithm can only be applied to the features with a value 'present' in at least some languages). The final feature set comprises therefore 171 features. Out of the 171 features, more than a half of the languages could be coded for 95% of features (162 features), around two-thirds of the languages could be coded for more than 78% of features (134 features) (see electronic supplementary material, figure S1 for the relationship between the amount of 'present', $D$ and rates, and electronic supplementary material, figure S2 for the relationship between missing data, $D$ and rates).

To investigate the evolutionary dynamics of features in more detail and to compare it across a relatively big number of features, I divided the features into 17 functional categories, five language domain categories and 10 part of speech categories. A short overview of the categorization and the main idea behind individual categories follow below, for the categorization of individual features, see electronic supplementary material, table S1.

---

[1]For the sake of simplicity, I use the term 'language' throughout instead of the more accurate term 'doculect', i.e. a language as it is described in the grammar. The information available in the grammar may be different from the current state of the language or variety holding the same name or different from the variety known to the reader.

The four chosen language domains (or levels) are: 'phonological shape' (14 features), 'word' (71 features), 'nominal phrase' (21 features), 'clause' (63 features) and 'other' (4 features). Features that target the form of the word (phonological shape) comprise vowel harmony (4 features), phonotactic constraints (3 features), voicing/aspiration distinctions in consonants (4 features), l/r distinction. The category 'clause' comprises features that have the whole clause as their scope. Most features have to do with phonologically free marking; in some features, there is variation, e.g. the feature on negation marking appearing clause-finally versus clause-initially: in many languages in the sample, negation is marked by a suffix on the verb, and, due to subject–object–verb (SOV) word order, it appears to be clause-final, although the negation marker is bound—the focus of the feature is on the position and not on the phonological boundness. Some of the feature sets included are: comparative construction (4 features), predicative possession (5 features), interrogation (7 features), negation (5 features). The category 'word' comprises features that target the word and where the presence of the feature is realized by a bound marker. The most prominent feature sets in this category include case marking, indexing, derivation, number and possession marking, morphological tense–aspect–mood (TAM) marking, and valency markers on verbs. The category 'NP' covers word order and agreement in the noun phrase. Features on the adpositions, articles and nominal conjunction are also included in this category. The features that could not be assigned to any of the above-mentioned categories were categorized as 'other'.

Parts of speech are a highly disputed topic in linguistics and it is by no means a trivial matter to assign words of one language to a particular part of speech, let alone when we deal with 60 languages at a time. For the current exploratory purposes, they nevertheless appear to be a useful proxy for explaining stability of particular features. The part of speech categories include features that could be described as targeting 'adjective' (5), 'article' (4), 'demonstrative' (3), 'noun' (19), 'particle' (12), 'pronoun' (15) and 'verb' (54). A significant number (15) of features concerns both nouns and pronouns, therefore a separate category ('noun/pronoun') appeared worthwhile. Features targeting the presence of pre- and post-positions and ideophones could not be conflated with any other part of speech and form their own category 'other' (3). A number of features (41) could not be assigned to a part of speech. These are mostly features that otherwise fall into the functional categories 'phonological distinctiveness' and 'word order' and language domain categories 'nominal phrase' and 'clause'.

The functional categories are: 'argument marking (core)' (10), 'argument marking (non-core)' (8), 'deixis' (15), 'derivation' (5), 'interrogation' (8), 'modification' (6), 'negation' (7), 'phonological distinctiveness' (14), 'possession' (11), 'quantification' (17), 'TAME+' (23), 'valency' (11), 'word order' (22) and 'other' (14). In the categories 'argument marking (core)' and 'argument marking (non-core)' both features on flagging (marking on the nouns and pronouns) and indexing (argument marking on the verbs) are included. 'Deixis' covers features on articles, pronouns (except case marking), and demonstratives. 'Derivation' includes features on deverbal and denominal derivation (action/state, agent and object derivation, diminutive and augmentative marking). 'Interrogation' covers features on the manner of expression of interrogation as well as position of the interrogation markers. 'Modification' includes features on the comparative construction and on adjectives acting as verbs. 'Negation' covers features on the negation of verbs and other types of predicators. 'Phonological distinctiveness' overlaps completely with the category 'phonological shape' from the language domain categorization. 'Possession' includes features on attributive (ways of expressing 'my goat') and predicative (ways of expressing 'I have a goat') possession. 'Quantification' spans over numeral systems, classifiers, nominal number marking and agreement in number in a nominal phrase. 'TAME+' covers both tense–aspect–mood–evidentiality marking (phonologically free and bound) and other non-derivational marking on or modification of verbs. 'Valency' includes features on causatives, applicatives, passives and other valency-related phenomena. 'Word order' spans from the order of components in the nominal phrase to order in the clause and the position of the relative clause according to the noun. Features that would require opening up small categories were grouped together in the category 'other'. Features on reduplication, verbal compounding, copula for predicate nominals, existential verb, ideophones, clause chaining, light verbs and others are included in this category.

## 2.2. Methods

To serve as the 'gold standard' reconstruction of the relationships between these languages, I constructed a phylogeny from the classification taxonomy from Glottolog (v. 4.2.1) [29] for each of the five language families in the dataset. Glottolog is an independent catalogue of language relationships and references.

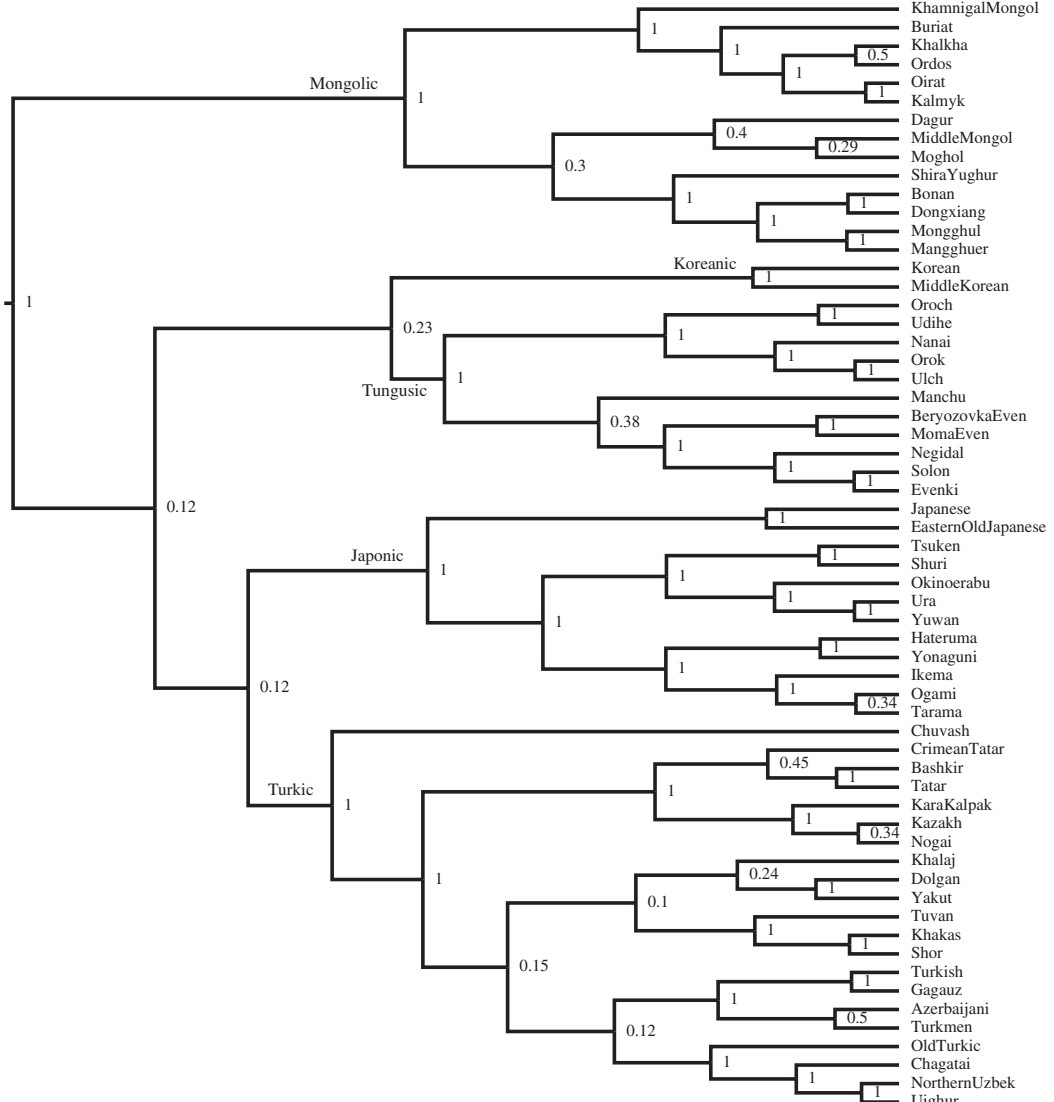

**Figure 2.** The maximum clade credibility tree: low probability on the node indicates either an agnostic view on the order of the splits or that a branch split into more than two further branches (i.e. a non-binary structure of the tree); high probability on the node (1) indicates the monophyletic constraint on the node according to the Glottolog (v. 4.2.1) [29] classification, except for the root probability value (1), which is an artefact of the fact that no languages, apart from Transeurasian, were included in the sample (i.e. there is no outgroup).

The subgroupings in the Glottolog classification were used to enforce monophyletic clade constraints (figure 2). The classifications are based on Johanson [31, pp. 161–162] for Turkic, Rybatzki [32, pp. 386–389] for Mongolic, and Pellard [33, pp. 5–8] for Japonic. The Tungusic classification is based on the three-branch proposal of the family by Doerfer [34] referred to by Whaley [35, p. 397].

I used BEAST (v. 2.5.1) [36] to build a data-free 'pseudo-posterior' of trees from this classification using a covarion model of evolution [37] and a relaxed clock model [38]. As there was no linguistic data beyond the tree topology, I ran this analysis for 10 000 000 generations, sampling a tree from the posterior every 1000 generations. This procedure provided a posterior probability distribution of the trees with each node in the Glottolog classification having a posterior probability of 1.0, while unresolved groupings were assigned low probabilities but—importantly—retaining the uncertainties in the language subgroupings. This allows me to adopt an agnostic view on the order of splits of branches. For example, according to Glottolog, the South Kipchak branch comprises three languages: Kara-Kalpak, Kazakh and Nogai. The branch itself has the probability 1.0, because these languages definitely belong to the same branch, but the probability of the Kazakh-Nogai branch is low, namely 0.35, which means that the clustering of Kazakh and Nogai together is arbitrary and Kazakh and Kara-Kalpak could have also belonged to one branch with similarly low probability. As there is no

information in Glottolog about relationships above the language family level like Transeurasian, the relationships between the language families are likewise arbitrary and all deep groupings are equally likely to appear in the posterior sample (note that posterior probabilities for the higher nodes are below 0.25).

To measure the phylogenetic signal, I use metric $D$ [39]. This metric calculates the sum of the differences between related branches in a tree. $D$ is the sum of sister clade differences across the tree and its values normally fall in the range between 0 and 1. In a trait that is strongly phylogenetically structured, sister languages will share the same value (and have no difference). If the trait is not phylogenetically patterned, then sister languages will have different values: the $D$ value will be high and the phylogenetic signal will be low. I computed the $D$ statistic on a sample of 1000 trees from the posterior probability distribution described above, using R (v. 4.0.1) [40] and the function *phylo.d* in the package *caper* (v. 1.0.1) [41].

To measure the evolutionary rate and reconstruct ancestral states, I used the Hidden Rates model as implemented in the function *corHMM* from the package *corHMM* (v. 2.4) [42]. The function allows to choose between two models: *ARD* (= all rates differ) and *ER* (= equal rates). As there are no strong grounds to assume that linguistic features are gained and lost at the same rate and as the differences between these rates will be particularly interesting for explaining linguistic diversity in future research, I chose the model *ARD*. The rate of gain ($0 \rightarrow 1$) is therefore allowed to be different from the rate of loss ($1 \rightarrow 0$). I set the root prior following Maddison *et al.* [43] and FitzJohn *et al.* [44]. The Hidden Rates model foresees in our case two rate classes, a fast one (F) and a slow one (S), and two possible values: 0 and 1. A feature value has an equal probability of belonging to a slow and to a fast rate class. Each observed feature value would therefore belong to one of the classes and have a particular value, e.g. 1F or 1S. Features belonging to different classes can potentially have different transition rates. According to this model, there are eight possible transition rates: 1F to 0F, 0F to 1F, 1S to 0S, 0S to 1S, 1F to 1S, 1S to 1F, 0F to 1S and 1F to 0S. The rates cannot be observed directly—the affiliation with different rate classes can only be derived from the states, therefore such a model is commonly known as a hidden Markov model [45, p. 726]. Ancestral states were estimated using marginal approach, which integrates over the states at other nodes and calculates the likelihood of state at each node [42].

I estimated Kendall's $\tau$ to measure the correlation between the phylogenetic signal and the evolutionary rate.

## 3. Results

Over a half of all features (63%) have a median $D$ value below 0.5 and 37% of features have a median $D$ value over 0.5. If we want to categorize the distribution of $D$ values with more precision, we can divide them into four categories, depending on the range the value falls into: $D < 0$ in overclumped features (46%), $D$ between 0 and 0.5 in features with a phylogenetic signal (17%), $D$ between 0.5 and 1 in randomly distributed features (23%) and $D > 1$ for overdispersed features (14%).

As for the rate, over half of the features are gained and lost at a slow rate: 68% for feature loss and 75% for feature gain. Only approximately one third of the features evolve at a fast rate: 32% for feature loss and 25% for feature gain.[2]

For a more fine-grained categorization, one could divide the features into three categories: below −0.5 'slow', between −0.5 and 0.5 'medium' and above 0.5 'fast' (table 1). We see that the group of features lost at a fast rate is bigger than the group of features gained at a fast rate and a reverse trend for the slow rate: there are more features gained at a slow rate than lost (see table 2 for the measures of centre and dispersion and figure 3 for the distribution of the $D$ and rate values; the $D$ and rate values for individual features can be found in electronic supplementary material, tables S2 and S3).

Despite the observations made by studies in evolutionary biology [20], I find a moderate positive correlation between the phylogenetic signal and the evolutionary rate (figure 4): $\tau$ for the rate of loss and gain is 0.51 and 0.5 respectively, $p$ values approximate 0, i.e. features with a high phylogenetic signal tend to evolve at a slower rate, while features with low phylogenetic signal tend to evolve at a fast rate. There are 58 (gain) and 34 (loss) features that have rate equal to 0 prior to $\log_{10}$

---

[2]I use the $\log_{10}$ transformed values of evolutionary rate for the further analysis of the results, because the transformation returns a normal distribution, which allows for a better visualization and interpretation of the distribution. I assume that features with the $\log_{10}$ value below 0 evolve rather slowly and those above 0 evolve rather fast. Since $\log_{10}$ transformation of 0 would produce infinite values, I replaced rates equal to 0 by a value close to 0 (0.0000000001) prior to $\log_{10}$ transformation.

**Table 1.** Evolutionary rate (proportion per category).

| category | gain | loss |
|---|---|---|
| slow | 0.65 | 0.47 |
| medium | 0.2 | 0.28 |
| fast | 0.15 | 0.25 |

**Table 2.** Phylogenetic signal and evolutionary rate, summarized based on the median value per tree per feature.

| metric | min | median | max | s.d. |
|---|---|---|---|---|
| $D$ | −4.97 | 0.04 | 2.34 | 1.43 |
| rate of loss | 0 | 0.33 | 99.98 | 17.35 |
| rate of gain | 0 | 0.14 | 95.59 | 15.15 |
| rate of loss ($\log_{10}$ transformed) | −10 | −0.48 | 2 | 4.08 |
| rate of gain ($\log_{10}$ transformed) | −10 | −0.85 | 1.98 | 4.74 |

transformation. These features are thus almost never lost or gained: they are either present in all languages (e.g. 56 out of 56 languages with enough information on the feature or 60 out of 60 languages, i.e. the whole sample) or absent in all but one or two languages (e.g. 'present' in 1 out of 60 or 2 out of 24 languages). For the distribution of the $D$ values across these features, see electronic supplementary material, figure S7.

We see an overall trend of an increase in evolutionary rate and in $D$ with an increase in language level up to the nominal phrase and then a slight decrease at the clause level (see figure 5 and table 3). There are no levels evolving at a fast rate (above 0) on average, apart from the features that could not be attributed to any level ('other'). The category 'other' is also distributed randomly on the phylogeny, alongside 'nominal phrase' (these are the only categories without a phylogenetic signal). Features operating on the levels 'word' and 'phonological shape' are lost at the lowest rate on average. The category 'word' maintains its dominant position in terms of slow rate of change also for feature loss, but is followed by the category 'nominal phrase'. In terms of phylogenetic signal, 'phonological shape' is the most overclumped category on average, followed by the category 'word'.

The majority of functional categories have median $D$ below 0.5 (i.e. the features have a phylogenetic signal or are overclumped), except for 'interrogation', 'quantification' and 'TAME+' (see figure 6 and table 4). Features belonging to the categories 'argument marking (non-core)' and 'derivation' are gained at the slowest rate on average, but the category 'argument marking (core)' is ahead of other functional categories in terms of rate of loss, followed by 'argument marking (non-core)', 'valency' and 'derivation'. In terms of phylogenetic signal, 'modification' is the most clumped functional category, followed by 'argument marking (core)' and 'valency'. The fastest changing and most overdispersed functional category is 'interrogation'. Functional categories 'deixis' and 'TAME+' follow 'interrogation' in the rate of loss and in the reverse order ('TAME+', then 'deixis') in the rate of gain. They also belong to the few overdispersed functional categories in terms of phylogenetic signal, alongside 'interrogation' and 'quantification'.

The most slowly evolving parts of speech are pronoun, noun and 'other' (i.e. adpositions and ideophones) (see figure 7 and table 5). 'Pronoun' is also the most clumped category in terms of phylogenetic signal, followed by the category 'noun/pronoun' (see §2 for clarification). There is no part of speech that would be lost fast: the median for all of them lies below 0. The relatively fast lost parts of speech are demonstrative, adjective, article and particle. The same four parts of speech are the only categories that have the rate of gain above zero. Apart from adjective ($D = 0.4$), these same categories are also overdispersed in terms of phylogenetic signal ($D > 0.5$), complemented by the category 'other'.

The reconstructability of features does not show remarkable variation across language families (figure 8): the proportions of features reconstructable as 'present' or 'absent' are similar enough in all five language families for a joint summary. About one fifth of the features (19–22%) in the feature set

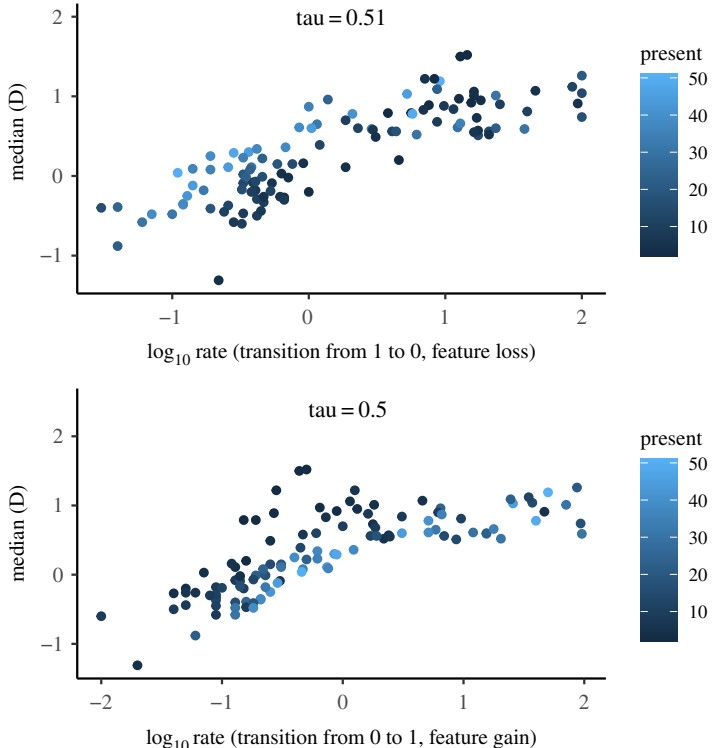

**Figure 3.** Distribution of phylogenetic signal (*D*) and evolutionary rate (transition from 0 to 1 and from 1 to 0) across 1000 trees and 171 features.

**Figure 4.** Correlation between phylogenetic signal and transition rates. Dark colour of the data points indicates features with most values 'absent' and light colour indicates features with most values 'present'. Features with the rates equal to zero prior to the $\log_{10}$ transformation (58 for feature gain and 34 for feature loss) are not included in the plot: these features are either present in all languages or absent in all but one or two languages.

can be reconstructed with 95% certainty, and around one third of the features (33–38%) can be reconstructed with 75% certainty of being present in the proto-language. About one fifth to one fourth (21–26%) of the features can be reconstructed with 95% certainty and almost a half (44–50%) with 75% certainty of being absent in the proto-language. Overall, there is only a small range of features (17–22%) that cannot be reconstructed as either 'present' or 'absent' (between 25% and 75% certainty in reconstruction as 'present') in the proto-language.

In order to make the reconstructability of features belonging to particular categories comparable across categories, I normalized the number of features reconstructable as 'present' (≥95%) by the number of features belonging to each category. This was necessary to eliminate the impact of some categories with a high number of features (e.g. 'word' (71), 'clause' (63), 'verb' (54), 'not assignable' (41), see §2 for details). There are some differences in the reconstruction across language domains and

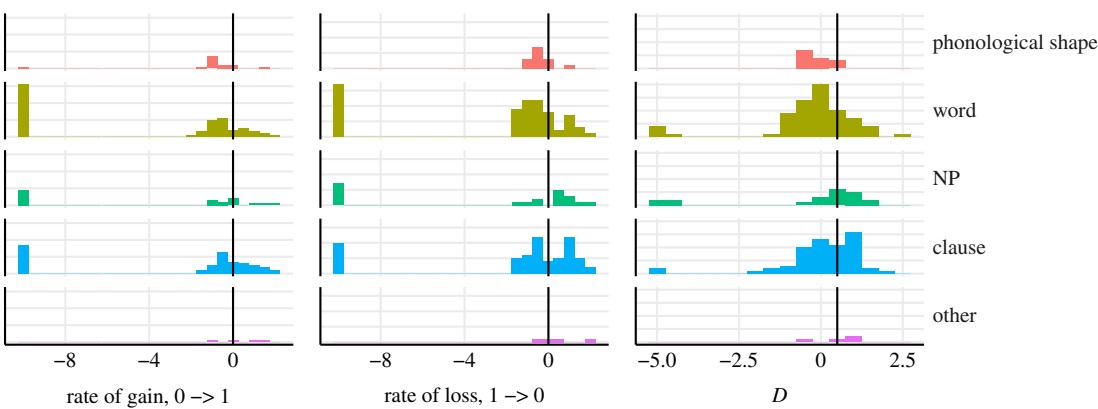

**Figure 5.** Feature loss and gain across different language levels. The vertical line marks the division between low and high log$_{10}$-transformed evolutionary rate and the division between phylogenetic signal and random distribution on the phylogeny.

**Table 3.** Phylogenetic signal and evolutionary rate across language levels.

| level | median rate (loss) | median rate (gain) | median D |
|---|---|---|---|
| phonological shape | −1.02 | −0.36 | −0.24 |
| word | −1.22 | −0.92 | −0.01 |
| NP | −0.89 | −0.39 | 0.57 |
| clause | −0.57 | −0.38 | 0.22 |
| other | 0.29 | 0.25 | 0.77 |

**Table 4.** Phylogenetic signal and evolutionary rate across functional categories.

| level | median rate (loss) | median rate (gain) | median D |
|---|---|---|---|
| argument marking (core) | −5.44 | −1.46 | −0.42 |
| argument marking (non-core) | −10 | −1.31 | −0.07 |
| deixis | −0.52 | 0.27 | 0.16 |
| derivation | −10 | −1.05 | 0.16 |
| interrogation | 0.68 | 0.97 | 0.84 |
| modification | −0.87 | −0.82 | −0.44 |
| negation | −0.57 | −0.59 | 0.08 |
| other | −5.52 | −0.85 | 0.04 |
| phonological distinctiveness | −1.02 | −0.36 | −0.24 |
| possession | −0.74 | −0.38 | 0.04 |
| quantification | −0.82 | −0.44 | 0.8 |
| TAME+ | −0.22 | 0.02 | 0.6 |
| valency | −1.05 | −1.15 | −0.29 |
| word order | −1.17 | −0.52 | −0.05 |

across language families (figure 9), but the overall trends overlap in most language families: the categories 'nominal phrase' and 'word' have the highest proportion of reconstructable features among level categories, 'pronoun' and 'other' (adpositions and ideophones) among parts of speech and 'derivation' among functional categories. Features in the category 'phonological shape' are better reconstructable for Mongolic and Tungusic languages than for the other families. There is a striking difference in the proportion of well-reconstructable features belonging to the categories 'verb' and

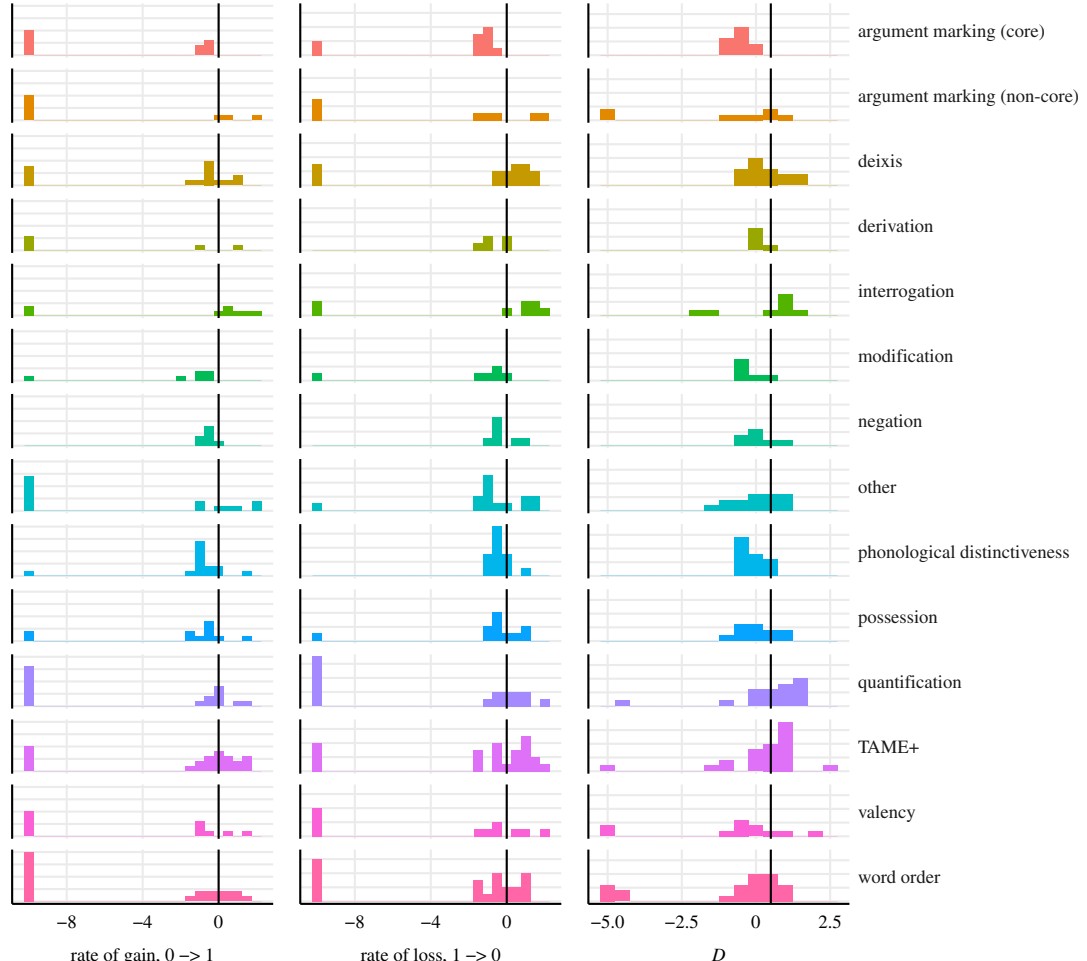

**Figure 6.** Feature loss and gain across different functional categories. The vertical line marks the division between low and high log$_{10}$-transformed evolutionary rate and the division between phylogenetic signal and random distribution on the phylogeny.

**Table 5.** Phylogenetic signal and evolutionary rate across parts of speech.

| part of speech | median rate (loss) | median rate (gain) | median D |
|---|---|---|---|
| adjective | −0.35 | 0.08 | 0.4 |
| article | −0.45 | 0.92 | 0.68 |
| demonstrative | −0.3 | 0.61 | 0.56 |
| not assignable | −1.05 | −0.43 | −0.08 |
| noun | −10 | −1 | 0.15 |
| noun/pronoun | −0.74 | −0.77 | −0.18 |
| other | −10 | −10 | 0.51 |
| particle | −0.58 | 0.62 | 0.93 |
| pronoun | −10 | −0.85 | −0.28 |
| verb | −0.7 | −0.59 | 0.11 |

'pronoun': 'pronoun' is the best reconstructable category in so-called Altaic languages, with 'verb' being rather moderately represented, but the distribution is the opposite for Japonic and Koreanic languages: here verbs are approximately equally well reconstructable as pronouns.

As might have been expected, the proportion of features that can be reconstructed to the proto-language with 95% probability for pairs of language families falls out slightly lower than the

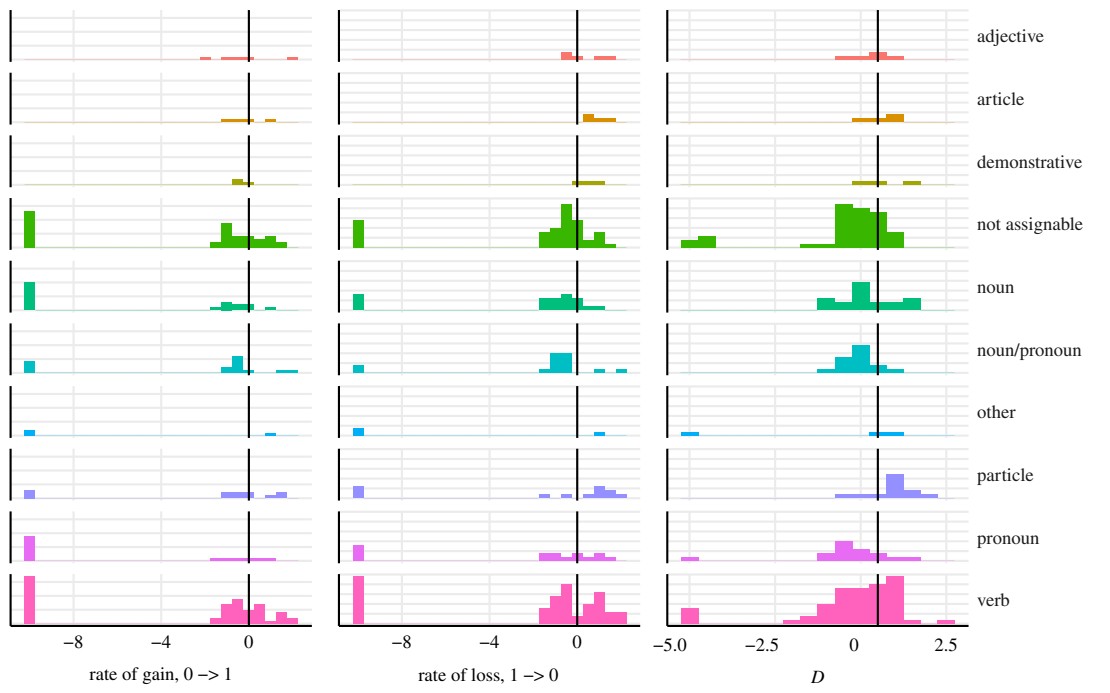

**Figure 7.** Feature loss and gain across different parts of speech. The vertical line marks the division between low and high $\log_{10}$ evolutionary rate and the division between phylogenetic signal and random distribution on the phylogeny.

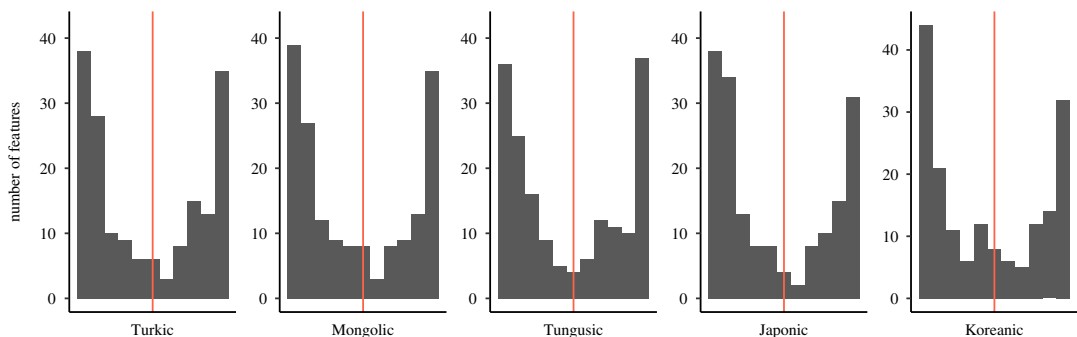

**Figure 8.** Reconstructed states per language family: $X$ axis from 0 (absent) to 1 (present), the red line indicates the point of highest uncertainty in reconstruction, 0.5.

proportion of features reconstructed for individual language families, e.g. we can reconstruct the same 33 features (19.14% of features) for Turkic and Mongolic together and 36 features for Turkic and Mongolic separately. A pairwise comparison of features reconstructable to proto-languages indicates that Turkic/ Tungusic and Mongolic/Tungusic pairs have the highest number of shared features (20.3%) that can be reconstructed with 95% probability as 'present' in the proto-language (see table 6 for the counts on other pairs). The tentative grouping of Japonic and Koreanic has the same amount of well-reconstructed features (17.4%) as Japonic/Turkic, Japonic/Tungusic and other pairs.

## 4. Discussion

In the terms used in this study, a feature is stable if it evolves at a relatively slow rate and has a high phylogenetic signal. The results have shown that 66% of the features in the dataset have a $D$ value below 0.5 and evolutionary rate below 0 (after the $\log_{10}$ transformation). Therefore, more than half of the features can be called relatively stable.

Why are some features more stable than others? Greenhill *et al.* [12, p. 4] and Arnold [46, p. 76] both independently propose the availability of a feature for reflection and analysis as a tentative explanation for its (in)stability. I see some support for this idea in my results: we can speculate that number marking

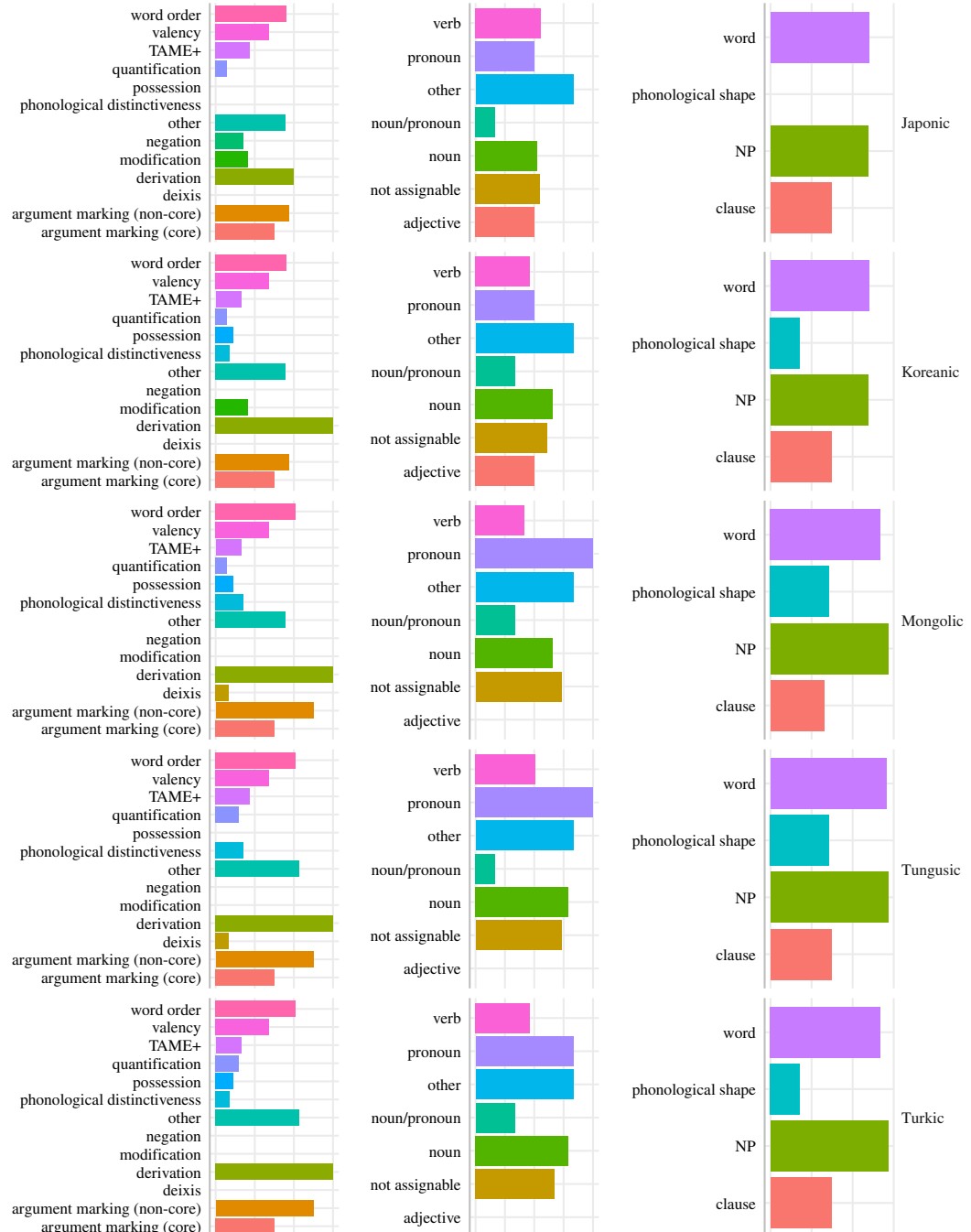

**Figure 9.** Proportion of features reconstructed with 0.95% certainty as 'present' across categories and language families. The number of features reconstructable per language family per category was normalized by the number of features belonging to each category.

**Table 6.** Overlaps in reconstructed features for pairs of language families at 95% probability of 'present', in %.

| language family | Turkic | Mongolic | Tungusic | Koreanic | Japonic |
|---|---|---|---|---|---|
| Turkic | X | 19.14 | 20.3 | 16.82 | 17.4 |
| Mongolic | 19.14 | X | 20.3 | 17.4 | 16.24 |
| Tungusic | 20.3 | 20.3 | X | 17.4 | 17.4 |
| Koreanic | 16.82 | 17.4 | 17.4 | X | 17.4 |
| Japonic | 17.4 | 16.24 | 17.4 | 17.4 | X |

is more analysable for speakers than derivation of nouns from verbs, core argument marking is less conscious than oblique argument marking, valency is less analysable than tense–aspect–mood marking. Interrogation probably needs most conscious processing by speakers compared with other categories and is thus more prone to change, be it as a result of a single innovation in the community of speakers or of a borrowing event. One could hypothesize that several ways of interrogation marking could exist in parallel in a language (possibly with one way dominant at a time) and the choice of marking could depend on the pragmatic situation (and the speaker could therefore decide upon the marking spontaneously). We could apply a similar explanation also to language levels: the higher the level, the more conscious are the speakers of their language use. It follows from the results that the features operating on the phonological and word level are the most stable. This trend does not extend on the nominal phrase and clause: here we see an opposite situation with 'clause' being more stable than 'NP', but it does not mean that the explanation of reflection cannot be applied here. On the contrary, the word order in the noun phrase, the use of adpositions, articles and conjunctions might be more available to analysis than the position of the negation marker, the verb, the way one expresses possession of the type 'I have a dog'. One should note that 'clause' appears relatively stable despite the fact that it includes features from the most unstable functional category, 'interrogation'. The ease of analysis of particular language categories by speakers would profit from a thorough investigation by other disciplines.

In some cases, variation in stability could also be explained by the frequency of use (on the relationship between frequency and stability, see Bybee & Thompson [47] and Diessel [48, p. 118]). We could apply this explanation to the parts of speech: pronouns and nouns appear to be the most stable and the most frequently used parts of speech. Articles, on the contrary, are neither obligatory nor frequent in the languages of the sample, therefore, as one might expect, they are rather unstable both in terms of rate and phylogenetic signal. This might be different for a different language family, where articles are obligatory. Following the logic of frequency of use, 'phonological distinctiveness' should be the most stable functional category, but it appears to be rather intermediate both in terms of rate and phylogenetic signal—we would need to develop an explanation for it in further research. In order to connect the stability of interrogation and other functional categories to the frequency of use, one would need a corpus of the languages in the sample (e.g. to compare the frequency of declarative and interrogative clauses and draw conclusions from the frequency distributions).

Another possible explanation is the areal spread of the feature. Word order in noun phrases appears to be identical in the whole area: all modifiers appear before the noun, and most of the languages in question have been in contact either with each other, or a neighbouring language, often Mandarin Chinese or Russian. It is therefore difficult to conclude that constituent order is *per se* a genealogically stable feature, if language contact cannot contribute to variation. As for clausal word order, it has some variation in the area: all Transeurasian languages have OV order, whereas influential neighbouring languages have VO order. In some Transeurasian languages, also VO order is possible, due to borrowing, but most of the languages could resist the influence of dominant neighbouring languages and retained OV order.

In order to better assess the results being generalizable to world's languages, it is worth comparing the stability of the features in my results to those of the previous studies. According to Nichols [49, p. 353], the most genetically[3] stable features are the alignment of head/dependent marking, inclusive/exclusive oppositions, gender, number oppositions in the noun, and detransitivation processes. For head/dependent marking, we consider features on attributive possession, indexing and core case marking (argument marking (core)). The results of this study support the findings of Nichols [49] in this respect: features belonging to the functional category 'argument marking (core)' ($D = -0.42$, $q01 = -1.46$, $q10 = -5.44$)[4] and features on indexing ($D = -0.35/-0.36$, $q01 = -0.68$, $q10 = -0.92$) are both lost and gained at a slow rate and are overclumped in terms of phylogenetic signal. Features on attributive possession evolve at a slow rate, but differ in their phylogenetic signal: marking of possession by a suffix on the possessed is overclumped ($D = -0.9$), on the possessor has a phylogenetic signal ($D = 0.04$) and marking the possessed with a prefix is extremely dispersed ($D = 1.04$, present only in one language, Hateruma). Inclusive/exclusive distinction (in the current language set: in pronouns only) is almost never gained ($q01 = -10$) and lost at a slow rate ($q10 = -0.42$), it is 'overclumped' in terms of phylogenetic signal ($D = -0.15$). There is no gender in the

---

[3]Terminology of Nichols [49] is preserved.

[4]$q01$ stands for the transition from 0 (absent) to 1 (present), or feature gain, $q10$ for the transition from 1 (present) to 0 (absent), or feature loss. The value $-10$ after the $\log_{10}$ transformation corresponds to the value 0 before the $\log_{10}$ transformation.

languages in question, therefore we can only say that gender is difficult to gain, despite neighbouring languages (e.g. Russian) having gender. The features in the functional category 'quantification', which includes more features than mentioned in Nichols (1993) [49], evolve at a relatively slow rate, but have no phylogenetic signal. Some features of interest here are: associative plural marker ($D = 0$, $q01 = -10$, $q10 = -1$), suppletion for number ($D = -1.24$, $q01 = -10$, $q10 = -10$) and non-phonological allomorphy of noun number markers ($D = 0.15$, $q01 = -0.6$, $q10 = -0.12$), which all evolve at a slow rate and have a phylogenetic signal. Plural marking on nouns ($D = 0.6$, $q01 = -0.07$, $q10 = -0.44$) also evolves at a slow rate, but is more dispersed than the other features on nominal number. As for the detransitivising processes, features on valency are gained and lost at a slow rate ($q01 = -1.15$, $q10 = -1.05$) and have a high phylogenetic signal ($D = -0.05$). Morphological passive marking evolves slowly and is overclumped in terms of phylogenetic signal ($D = -0.29$, $q01 = -10$, $q10 = -1.15$). Even if these are not the only features with a high phylogenetic signal and evolving at a slow rate, the results of the current study do not contradict the findings of Nichols [49] in most cases.

In her 1992 book [16, p. 167], Nichols mentioned word order as being very genetically unstable. This finding is only partially supported by my results: the category 'word order' as a functional category evolves at a slow rate ($q01 = -0.52$, $q10 = -1.17$) and has a high phylogenetic signal ($D = -0.05$), but the stability varies for particular orders of constituents (OV is more stable than VO). Nichols [49, p. 353] classifies clause word order as areally stable, and this is definitely true for the current language sample: the word order is verb-final in all languages in the sample ($D = -4.95$, $q01 = -10$, $q10 = -10$), with four languages (Gagauz, Khalaj, Beryozovka Even and Moghol) also allowing verb-medial word order due to borrowing ($D = 1.07$, $q01 = 0.06$, $q10 = 1.21$). It is difficult to conclude that verb-final word order is more stable than verb-medial word order: in a language sample, where 60 out of 60 languages have verb-final word order, this feature has not changed, but this might well be the case for verb-medial word order in an area or a language family, where this particular order dominates.

Greenhill *et al.* [12] quantify the stability of structural features in terms of evolutionary rate. Based on a sample of 81 Austronesian languages, this study sets apart the following features as being particularly stable: inclusive versus exclusive distinctions, gender distinction in third person only in pronouns, tone, future marking on the verb, conflation of categories (e.g. alignment, conflation of second and third persons in non-singular numbers), which mostly overlap with those of Nichols [49]. According to my results, morphological future marking has a phylogenetic signal and is lost and gained at a slow rate ($D = 0.11$, $q01 = -0.51$, $q10 = -0.42$). Gender in third person pronouns ($D = -0.43$, $q01 = -10$, $q10 = -10$) is almost never distinguished in the languages in question, apart from Japanese, where pronouns are generally omitted (and third person pronouns the more so). There is no data on the conflation of second and third persons in non-singular numbers available, and this is not relevant for the area in question. The current results thus go in line with the findings of Greenhill *et al.* [12].

One of the conclusions of a recent study on the evolutionary rate in structural features based on Indo-European languages [24] is that morphological features evolve slower than syntactical features. The current study provides evidence in support of this conclusion: the level 'word' evolves at the slowest rate among all level categories and takes in the second position in terms of phylogenetic signal, giving way only to 'phonological shape'.

Since we see support of the presented results in previous studies, which did not focus on the same region, and since this study covers five language families (albeit with hypothesized genealogical relationships or at least forming a sprachbund), we can say that stability patterns of different areas of grammar might have a cross-linguistic component. Nevertheless, we can only draw conclusions on the stability of particular features for one unit at a time, be it a language family or an area, because we can only measure features that are present in the given family or area. We cannot fully compare these results with those for Indo-European or Austronesian, because there are typical Austronesian, Indo-European and 'Transeurasian' features.

Already, at the stage of data collection, it becomes obvious that the area is very homogeneous: many features in the questionnaire are either invariable or deviate for very few languages (whether present or absent): 118 features are present in 50 languages (out of 60) or more, 53 features of the initial 224 features appeared to not to be present in the area. This can be explained by the nature of the questionnaire: at the core of the feature set from Grambank is the selection of features exhibiting some variation in the languages of Island Melanesia [6]. Therefore, features interesting for that area are uninformative for Northern Eurasia and were discarded in further analysis.

This development is not new in linguistics: there were several adjustments to the basic vocabulary list after it was shown that not all items on the lists are universal and present in all languages, as they were originally claimed to be. Some languages appeared e.g. to lack words for 'snow', 'ice', 'freeze' and 'sea'

[50,51]. The same way as these words typical for cold regions of the Earth tend to be stable in the languages, where they are present, some features are typical and stable in one part of the world, but completely absent in the other part of the world—and thus irrelevant for the studies on stability of structural features in that region.

We could think of stability not as a universal phenomenon, but as a trend that depends on the area and genealogical affiliation: one could expect language families with similar typological profiles (e.g. more synthetic or more analytic) to show similar stability patterns in their grammars, e.g. that morphology will tend to be stable if the language makes use of extensive morphological marking. If the language rather uses free marking more often than bound marking, then one might hypothesize that this marking will be more stable due to its higher frequency. The current study is thus a step towards a list of universally stable structural features (if it were ever to be determined), but not the final destination in the construction thereof: it provides ground for testing new hypotheses for stability patterns in other language families.

Given the correlation between the rate and the phylogenetic signal mentioned in §3, there is often an overlap between the rate of loss, gain and phylogenetic signal across categories. Can we conclude from this that it is inessential to investigate these two dynamics simultaneously and one metric would have been sufficient? It appears that we would lose information if we only considered either of the two: if we only accounted for the rate of loss, we would conclude that articles are definitely stable, because they are difficult to lose, but considering also the rate of gain we see that they are far more easy to gain and do not have a phylogenetic signal, but are rather distributed randomly on the tree. Therefore, a straightforward conclusion that articles are clearly stable would be misleading. Overall, there is a discrepancy between the phylogenetic signal and evolutionary rate in 40 features, i.e. some features with high $D$ evolve slowly and some features with low $D$ evolve fast. The method used in this study allows us to get a more complete picture of the evolutionary dynamics of structural features and prevents us from making precipitous conclusions.

We have seen that more than a half of the structural features bear a high phylogenetic signal and evolve at a slow rate. If the features are preserved relatively well, can we reconstruct them to the proto-language level? It might be interesting to compare the reconstructability of features across language families to see if there are family-specific trends, e.g. is there an especially innovative language family, where very few features can be reliably reconstructed, or an overly conservative language family, where most of the features can be traced back to the proto-language? We could also determine the features that can be reconstructed as 'present' to the proto-language level across all five families.

Ancestral state reconstruction can be performed most reliably if most members of the family are sampled. Unfortunately, we cannot know how many languages once belonged to the five language families in the sample, but we can assume that the current number of languages in these families is only a small fraction of the languages once spoken in the area, especially given that nowadays most Tungusic languages are severely endangered, most Mongolic, Turkic and Japonic (Ryukyuan) languages do not enjoy high prestige, or speakers are discouraged from using their native languages, there are only two Koreanic languages in the sample, one of which is merely the older stage of Modern Korean. This is a limitation of the study, which we have to bear in mind when interpreting the results, but there is no known possibility at the moment to account for it.

As for the question on the conservatism in particular language families, I find that the proportion of features that can be reconstructed as 'present' or 'absent' at the proto-language level does not vary substantially across language families (within 3% for 'present' and 5% for 'absent' at 95% probability), i.e. any of the five proto-languages can be reconstructed equally well. If we lower the probability boundary to 75%, then Turkic with 38% of features reconstructable as 'present' stands out slightly as being more conservative than other language families with their 33.3%–34.5% of reconstructable features.

In §3, I discussed categories of features that can be reconstructed well ($\geq$ 95%) for each of the language families, approximately one fifth of the features. It is not surprising that most categories that are best reconstructable in all language families also are the categories that evolve at the slowest rate in most cases ('argument marking non-core' and 'derivation' among functional categories, 'pronoun' among parts of speech). We see an unexpected pattern in the language levels: we would assume that relatively few features operating on the level of the noun phrase should be reconstructed well to the proto-language level, because this is the fastest evolving category, but in terms of reconstruction, the category 'noun phrase' can well compete with 'word'.

Apart from well-reconstructable features, there is one fifth of features that can be reconstructed rather poorly (around 0.5 probability of being 'present'). Poor reconstruction of features is partly indirectly due to an unequal distribution of presence and absence across branches, e.g. present in 30 languages out of

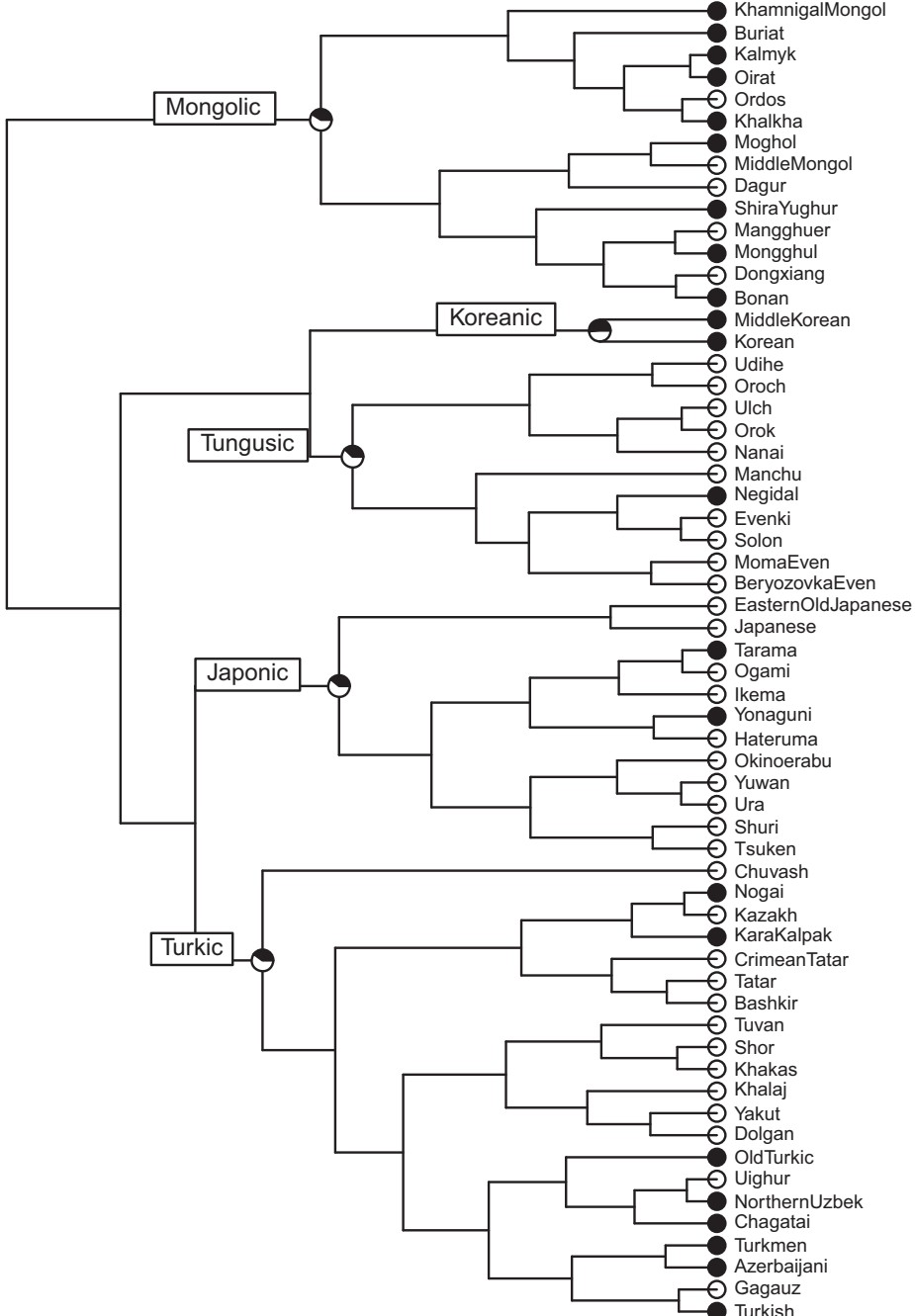

**Figure 10.** Distribution of the feature TE027 on the tree. The circles on the tips indicate feature values for individual languages and the circles on the nodes the reconstructed ancestral states for five proto-languages: 'present' in black, 'absent' in white.

60 languages. Features with such an unequal distribution comprise, among others, marking of comitative and conjunction, productive plural marking, future tense marking on the verb, tense marking by an auxiliary verb, etc. There might also be a joint effect of a random distribution on the tree and a high amount of missing data. Such features include a class of patient-labile verbs, an inclusory construction, a morpho-syntactic distinction between controlled vs. uncontrolled events, etc. These features are codable as 'present' only for several languages and are either absent in other languages or there is no information on their presence available.[5] Therefore, if a feature has only 50/50 of '1' and

---

[5]The decision on coding the feature as absent versus unknown is often made based on the overall quality of the grammar description work. For example, if there is no information on the feature in a grammar sketch of 30 pages, then the feature usually receives a '?'. If the feature is not mentioned in an extensive 600 pages grammar, then the feature is most probably absent.

'?' values inside a family, the reconstruction will not be only 1—it will incorporate the uncertainty by allowing some chance of the feature being '0' at the node level: the model assumes polymorphic '0/1' values at the tips with '?'. This way, we are not making false assumptions of full reconstructability of a feature if there are several '1' values and much missing data.

From the computational side, poor reconstruction is an effect of the interaction between the evolutionary rate and the values at the tips: if the model itself is not sure about the ancestral state, like the 'equal rates' model, it lets more information flow into the ancestral state. With the 'all rates differ' model, the reconstruction is less susceptible to slight differences in tip values [52, p. 476]. We will take feature TE027 'Can 1PL marker be augmented by a collective plural marker?' as an illustrative example for the interaction between the model assumptions and the tips (see figure 10 for the distribution of the feature on the tree and the reconstructed ancestral states). It is probable that in three language families (Turkic, Tungusic and Japonic) the feature was innovated on some branches, whereas it is present in both Koreanic and most Mongolic languages and we would intuitively think it was lost in some of the Mongolic languages. Nevertheless, this feature is reconstructed with high certainty in none of the families: even in Mongolic and Koreanic the probability of this feature being '1' at the proto-language level is not higher than 0.51.

Why is the feature TE027 reconstructed for Proto–Koreanic almost as a 50/50 chance of being present if both Middle Korean and Modern Korean have it and why is its probability of being 'present' so low for Mongolic, even though most Mongolic languages seem to have an additional plural marker on the 1PL personal pronoun? It is most probably due to the interaction effect between the evolutionary rate and the tip values described above: the ancestral state reconstruction is informed about the evolutionary rate of this feature, which is relatively high, and the chosen model is *ARD*, therefore the impact of the tip values is moderate. We get thus a rather high uncertainty in the reconstructed node value for all families. In this case, the model is not necessarily useless: it is warning us to reconstruct this fast evolving feature with great care.

The combination of reconstructed states and the rate of change of particular features can allow further research to contextualize the rates in time, if there is enough information on the age of the proto-language. For example, the age of Proto-Turkic was estimated to be around 2100 years before present [5]. In the first step, one extracts the features reconstructed as 'present' in Proto–Turkic (95% probability of being 'present' or higher). In the next step, one calculates the distance between Proto-Turkic and its children languages on these traits. This procedure provides us with a number of differences between Proto–Turkic and each child language.

# 5. Conclusion

Structural features as another tool for gaining information on the relationships between languages are gaining importance in the field of historical linguistics. In order for structural features to be competitive, they need to have a comparable performance for reconstructing ancient relationships (i.e. stability) as basic vocabulary does. We can test this performance by analysing the dynamics of change in structural features, best measured as the phylogenetic signal and the rate of change. Even though the study presents results on five language families (Turkic, Tungusic, Mongolic, Japonic and Koreanic), the type of data and the transparent methodology make it possible for the results to be replicated on other language families to obtain a cross-linguistically stable set of structural features. Extracting the features with a high phylogenetic signal and evolving at a slow rate would enable us to compare the performance of the most stable vocabulary with the most stable structural features instead of a random set of features. This feature set can then be applied for testing hypotheses about language history on relatively equal terms with basic vocabulary.

Data accessibility. The scripts used in the study are available in the following GitHub repository: https://github.com/NataliiaHue/stability. The data are provided in electronic supplementary material [53]. The complete database of structural features with sources, comments and examples is available on https://doi.org/10.5281/zenodo.5720838 and the code for the analysis and other materials on https://doi.org/10.5281/zenodo.6257956.

Competing interests. I declare I have no competing interests.

Funding. Open access funding provided by the Max Planck Society.

The research leading to these results has received funding from the European Research Council (ERC) under the European Union's Horizon 2020 research and innovation programme (grant agreement no. 646612) granted to Martine Robbeets and from the Department of Linguistic and Cultural Evolution, Max Planck Institute for the Science of Human History.

**Acknowledgements.** I am grateful to Simon Greenhill for his valuable input on the implementation of analysis, targeted problem solving and his continued general support. I also thank Volker Gast and Jakob Lesage for their insightful comments on the paper, Robert Forkel and Christoph Rzymski for their help in data formatting and curation and James Boyko for his help with the corHMM package.

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
