## [Peer Review File · Royal Society Open Science]

Review History

RSOS-211252.R0 (Original submission)

Review form: Reviewer 1

Is the manuscript scientifically sound in its present form?

Yes

Are the interpretations and conclusions justified by the results?

Yes

Is the language acceptable?

Yes

Do you have any ethical concerns with this paper?

No

Have you any concerns about statistical analyses in this paper?

No

Recommendation?

Accept with minor revision (please list in comments)

Comments to the Author(s)

The paper presents a statistical investigation of the diachronic stability and phylogenetic signal of a range of typological features from the GramBank database, evaluated against a sizeable collection of Northern Eurasian languages.

The study is executed in a professional manner. The results are well-justified and relevant. The paper devotes adequate attention to the linguistic interpretation of the findings. For these reasons, I recommend publication.

There are a few minor comments that I would like to bring to the author's attention:

- The numerical values of the rates of change are hard to interpret. I assume that the unit here is just the average rate, but I am not sure. Anyway, interpretation of the results would be helped considerably if these rates would be set in relation to real time. This is admittedly difficult to achieve, but giving, e.g., the age of Proto-Turkic in average numbers of mutations would be very useful to relate these numbers to the real world.
- When explaining the D statistic, it is said that a value of 0 corresponds to evolution "according to Brownian motion" (page 6). This is confusing because it applies to binary features, while Brownian motion is a dynamics of continuous features. Please clarify.
- page 17, lines 40/41: It is said that "[t]he 'all rates different' model obviously allows the evolutionary rate to vary in different parts of the tree". I believe the ARD model allows gain rates and loss rates to be different, but both rates are still constant across the tree. Please check.

Review form: Reviewer 2

Is the manuscript scientifically sound in its present form?

No

Are the interpretations and conclusions justified by the results?

No

Is the language acceptable?

Yes

Do you have any ethical concerns with this paper?

No

Have you any concerns about statistical analyses in this paper?

No

Recommendation?

Major revision is needed (please make suggestions in comments)

Comments to the Author(s)

This paper makes contributions to our understanding of the relationship between phylogenetic signal and transition rates for structural linguistic features and the stability of various parts of the grammar in 5 language families, and presents an approach for investigating the stability of

structural features. Importantly, it reports a correlation between phylogenetic signal and rate of change in structural language features.

As currently written, this paper would seem to be of most interest to linguists interested in theories of relatedness among the five study families (e.g. Transeurasian or Altaic studies). There is promise here for this to be framed more generally as a blueprint for investigating stability of typological characteristics in the world's languages. However, the current manuscript foregrounds comparisons of the 5 families in the study and the big, vague, still-open question of whether typological features are stable over time and places less emphasis on the patterns of stability across areas of grammar that its approach has the ability to probe. The paper offers some useful information about similarities between families of northern Asia and a bit of evidence that tends to support prior research on what grammatical features might be relatively stable.

While this paper is a step in the direction of assessing how structural features may be used in inferring language phylogenies, as discussed in the conclusions (p. 16-17), it might be of more interest to scholars working outside of the study region/language families if the main text focused more clearly on differences in stability across areas of the grammar and made a better case for the results of this study being a) indicative of genealogical rather than areal signals, and b) generalizable to the world's languages.

Regarding the extent to which this study's insights are generalizable outside of the 5 language families it focuses on, the author reflects on p. 12-13 about the problem of biases introduced by studies focused on particular language areas or families, especially w.r.t. the source of some of the linguistic data used in this paper. This raises, but does not directly address, the bigger question of whether the results of this study are truly generalizable or whether they too reflect regional patterns that may not adequately represent general processes of language evolution. What does the stability of features in these five families tell us about stability in languages of the world?

Some of the links between ostensibly "Transeurasian" languages have been hypothesized to reflect language contact, rather than mutual inheritance from a common ancestor. This paper's introduction (p. 2, lines 47-48) proposes that testing the stability of features in a set of families whose relationships may be shaped by both contact and genealogical inheritance offers an advantage, but it's not made clear what that advantage is. Contact is little discussed in the remainder of the paper but is highlighted again in the final sentence of the conclusions. That passage (p. 17, lines 12-14) does not explain how the possibility of contact-based influence between languages in the sample confers an advantage in this research, but rather it casts some doubt on whether the findings actually demonstrate phylogenetic stability or whether they might instead tell us something about areal relationships. It would seem that focusing on a set of families thought to have been influenced by language contact might actually be a disadvantage for demonstrating phylogenetic stability of features. I don't think any historical linguist who is skeptical about the use of structural/typological features in inferring phylogenetic relationships would be reassured by the Conclusion section of this manuscript, and I believe that readers interested in language or cultural evolution outside of Transeurasian would like to see more discussion of possible areal influences throughout the interpretation of results (e.g. in the discussion of reconstruction of individual features on p. 15).

Finally, I believe that presenting the findings regarding stability profiles of different sets of structural features more prominently would generate more interest across a broad audience that may not have much familiarity with Transeurasian languages. The feature categories are not introduced until the discussion section, and even then the justification for splitting the feature set into these particular functional categories and levels of grammar used is not made clear. Until

this is discussed, reader interpretation of results is hampered by a lack of clarity about the nature of the data being analyzed.

Some other smaller points follow:

There is a long discussion on p.6 (lines 32-59) of phylogenetic signal D values that might be better presented in tabular format than prose.

The discussion of D value ranges on p. 5, lines 56-58 repeats information already conveyed on p. 4.

On p. 15, the potential for missing data to impact results is discussed. It would be useful to see an earlier mention of how much data is missing in this dataset. This is also important for a reader's understanding of the results.

Figure 2 labels internal nodes only with probabilities. I recommend also including family labels for the nodes that represent the proto-languages of the 5 families (as in Fig. 10).

In several plots you show information about transition rates for feature gain and loss. These are labeled with "0->1"/"0 to1" and "1->0"/"1 to 0" respectively, but it might be useful for a general audience to also include "(gain)" and "(loss)" in those figure labels (Fig. 3, 4, 6, 7, 8).

There are a number of typos throughout that need to be found and fixed. For example, p. 9, line 31 repeats the word "the" and the caption for Figure 5 spells "present" as "present". I recommend another round of editing for typos and minor grammatical matters.

Decision letter (RSOS-211252.R0)

Dear Mrs Hübler

The Editors assigned to your paper RSOS-211252 "Phylogenetic signal and rate of evolutionary change in language structures" have now received comments from reviewers and would like you to revise the paper in accordance with the reviewer comments and any comments from the Editors. Please note this decision does not guarantee eventual acceptance.

Please submit your revised manuscript and required files (see below) no later than 21 days from today's (ie 14-Oct-2021) date. Note: the ScholarOne system will 'lock' if submission of the revision

is attempted 21 or more days after the deadline. If you do not think you will be able to meet this deadline please contact the editorial office immediately.

on behalf of Professor Kevin Padian (Subject Editor)
openscience@royalsociety.org

Editor Comments to the Author:

Thanks for your submission. As you see the reviews are a bit mixed but generally positive. Both reviewers have methodological questions and some concerns about interpretations. Please address these in your revision. Best wishes.

Reviewer comments to Author:

Reviewer: 1

Comments to the Author(s)

The paper presents a statistical investigation of the diachronic stability and phylogenetic signal of a range of typological features from the GramBank database, evaluated against a sizeable collection of Northern Eurasian languages.

The study is executed in a professional manner. The results are well-justified and relevant. The paper devotes adequate attention to the linguistic interpretation of the findings. For these reasons, I recommend publication.

There are a few minor comments that I would like to bring to the author's attention:

- The numerical values of the rates of change are hard to interpret. I assume that the unit here is just the average rate, but I am not sure. Anyway, interpretation of the results would be helped considerably if these rates would be set in relation to real time. This is admittedly difficult to achieve, but giving, e.g., the age of Proto-Turkic in average numbers of mutations would be very useful to relate these numbers to the real world.
- When explaining the D statistic, it is said that a value of 0 corresponds to evolution "according to Brownian motion" (page 6). This is confusing because it applies to binary features, while Brownian motion is a dynamics of continuous features. Please clarify.
- page 17, lines 40/41: It is said that "[t]he 'all rates different' model obviously allows the evolutionary rate to vary in different parts of the tree". I believe the ARD model allows gain rates and loss rates to be different, but both rates are still constant across the tree. Please check.

Reviewer: 2

Comments to the Author(s)

This paper makes contributions to our understanding of the relationship between phylogenetic signal and transition rates for structural linguistic features and the stability of various parts of the grammar in 5 language families, and presents an approach for investigating the stability of structural features. Importantly, it reports a correlation between phylogenetic signal and rate of change in structural language features.

As currently written, this paper would seem to be of most interest to linguists interested in theories of relatedness among the five study families (e.g. Transeurasian or Altaic studies). There is promise here for this to be framed more generally as a blueprint for investigating stability of typological characteristics in the world's languages. However, the current manuscript foregrounds comparisons of the 5 families in the study and the big, vague, still-open question of whether typological features are stable over time and places less emphasis on the patterns of stability across areas of grammar that its approach has the ability to probe. The paper offers some useful information about similarities between families of northern Asia and a bit of evidence that tends to support prior research on what grammatical features might be relatively stable.

While this paper is a step in the direction of assessing how structural features may be used in inferring language phylogenies, as discussed in the conclusions (p. 16-17), it might be of more interest to scholars working outside of the study region/language families if the main text focused more clearly on differences in stability across areas of the grammar and made a better case for the results of this study being a) indicative of genealogical rather than areal signals, and b) generalizable to the world's languages.

Regarding the extent to which this study's insights are generalizable outside of the 5 language families it focuses on, the author reflects on p. 12-13 about the problem of biases introduced by studies focused on particular language areas or families, especially w.r.t. the source of some of the linguistic data used in this paper. This raises, but does not directly address, the bigger question of whether the results of this study are truly generalizable or whether they too reflect regional patterns that may not adequately represent general processes of language evolution. What does the stability of features in these five families tell us about stability in languages of the world?

Some of the links between ostensibly "Transeurasian" languages have been hypothesized to reflect language contact, rather than mutual inheritance from a common ancestor. This paper's introduction (p. 2, lines 47-48) proposes that testing the stability of features in a set of families whose relationships may be shaped by both contact and genealogical inheritance offers an advantage, but it's not made clear what that advantage is. Contact is little discussed in the remainder of the paper but is highlighted again in the final sentence of the conclusions. That passage (p. 17, lines 12-14) does not explain how the possibility of contact-based influence between languages in the sample confers an advantage in this research, but rather it casts some doubt on whether the findings actually demonstrate phylogenetic stability or whether they might instead tell us something about areal relationships. It would seem that focusing on a set of families thought to have been influenced by language contact might actually be a disadvantage for demonstrating phylogenetic stability of features. I don't think any historical linguist who is skeptical about the use of structural/typological features in inferring phylogenetic relationships would be reassured by the Conclusion section of this manuscript, and I believe that readers interested in language or cultural evolution outside of Transeurasian would like to see more discussion of possible areal influences throughout the interpretation of results (e.g. in the discussion of reconstruction of individual features on p. 15).

Finally, I believe that presenting the findings regarding stability profiles of different sets of structural features more prominently would generate more interest across a broad audience that may not have much familiarity with Transeurasian languages. The feature categories are not introduced until the discussion section, and even then the justification for splitting the feature set into these particular functional categories and levels of grammar used is not made clear. Until this is discussed, reader interpretation of results is hampered by a lack of clarity about the nature of the data being analyzed.

Some other smaller points follow:

There is a long discussion on p.6 (lines 32-59) of phylogenetic signal D values that might be better presented in tabular format than prose.

The discussion of D value ranges on p. 5, lines 56-58 repeats information already conveyed on p. 4.

On p. 15, the potential for missing data to impact results is discussed. It would be useful to see an earlier mention of how much data is missing in this dataset. This is also important for a reader's understanding of the results.

Figure 2 labels internal nodes only with probabilities. I recommend also including family labels for the nodes that represent the proto-languages of the 5 families (as in Fig. 10).

In several plots you show information about transition rates for feature gain and loss. These are labeled with "0->1"/"0 to1" and "1->0"/"1 to 0" respectively, but it might be useful for a general audience to also include "(gain)" and "(loss)" in those figure labels (Fig. 3, 4, 6, 7, 8).

There are a number of typos throughout that need to be found and fixed. For example, p. 9, line 31 repeats the word "the" and the caption for Figure 5 spells "present" as "present". I recommend another round of editing for typos and minor grammatical matters.

===PREPARING YOUR MANUSCRIPT===

===PREPARING YOUR REVISION IN SCHOLARONE===

Author's Response to Decision Letter for (RSOS-211252.R0)

See Appendix A.

RSOS-211252.R1 (Revision)

Review form: Reviewer 3

Is the manuscript scientifically sound in its present form?

No

Are the interpretations and conclusions justified by the results?

Yes

Is the language acceptable?

Yes

Do you have any ethical concerns with this paper?

No

Have you any concerns about statistical analyses in this paper?

Yes

Recommendation?

Accept with minor revision (please list in comments)

Comments to the Author(s)

This is a great paper and it will move the field ahead. There are some remaining issues though that need to be taken care of before the results can be considered sufficiently robust and clear:

1. The methods presuppose the existence of the phylogeny that they are used on, i.e. that all languages trace back to a common ancestor. Since this is heavily disputed in the case of Transeurasian and because alternative forces might impact stability (areal spread, as insightfully

discussed by the author in the discussion), it is critical to know whether the results are robust against this assumption. A straightforward way out is to perform a sensitivity analysis on trees that do not assume Transeurasian. (Areal spread for example would predict unequal rate variation across families: less when exposed to more spread, more when exposed to less spread.)

2. I am missing any model comparison, or at least model criticism. A standard approach is to compare the AIC across models (easy to compute for HMM models), with different numbers of rate categories; at least we need to know how a model with no rate variation compares to a model with rate variation. This is especially important to assess rate variation across families (see Issue #1)

3. Just to be sure I understand the "pseudo-posterior" trees: were the monophyletic constraints all and only those who are then annotated with an MCC post. prob. = 1? If so, please say so; if not, please explain in the text where the constraints were applied and what else might have led to post. prob. = 1. Also, perhaps I missed it, but where was the uncertainty used and reported for rates and D? Please make this explicit.

4. Since the trees are not timed, it is not appropriate to talk about "stable" in an absolute sense, only in the sense of "feature f_i is more/less stable than f_j ". While it looks like the author is fully aware of this, care should be taken to remove any remaining ambiguity here (e.g. p 7, opening paragraph of discussion).

5. The D metric assumes an expected signal based on a threshold model. Since such models are not standard in linguistics, this needs to be justified more explicitly, especially since the underlying assumption about a latent liability is not self-evident for language data. Also the discussion section should make clear to what extent the results depend on the threshold model assumptions; some readers might object to assumptions. Could the assumption be relaxed?

6. The correlations in Fig 4 need more attention. At least we need a separate analysis for log rates near 0. For such rates one would predict more values of D below 0 (or at least below .5) than above 0. This is hard to judge from the plot and already a quick histogram or density plot would help; better yet a formal analysis (e.g. through a hurdle or zero-inflation model)

7. For easy replicability, please turn all absolute paths to relative paths in the scripts and provide a table of contents in the readme, with a brief explanation what each script does and what each directory contains.

8. It was impossible for me to keep track of which legend belongs to which figure in the Supp. Materials; the tables appeared as latex code printed to PDF. Unreadable. Also, it would help if the captions explained briefly what the figures/tables are supposed to show and what went into them (ideally with a pointer to the code that generated the figure/table).

Decision letter (RSOS-211252.R1)

Dear Mrs Hübler

On behalf of the Editors, we are pleased to inform you that your Manuscript RSOS-211252.R1 "Phylogenetic signal and rate of evolutionary change in language structures" has been accepted for publication in Royal Society Open Science subject to minor revision in accordance with the referees' reports. Please find the referees' comments along with any feedback from the Editors below my signature.

Please submit your revised manuscript and required files (see below) no later than 7 days from today's (ie 31-Jan-2022) date. Note: the ScholarOne system will 'lock' if submission of the revision is attempted 7 or more days after the deadline. If you do not think you will be able to meet this deadline please contact the editorial office immediately.

on behalf of Prof Kevin Padian (Subject Editor)
openscience@royalsociety.org

Editor Comments:

Thanks for your revision, which seems to have satisfied most of the reviewers' concerns. There seem to be some remaining points to clarify, and if you require more time for this please let our Editorial Office know. Best wishes.

Reviewer comments to Author:

Reviewer: 3

Comments to the Author(s)

This is a great paper and it will move the field ahead. There are some remaining issues though that need to be taken care of before the results can be considered sufficiently robust and clear:

1. The methods presuppose the existence of the phylogeny that they are used on, i.e. that all languages trace back to a common ancestor. Since this is heavily disputed in the case of Transeurasian and because alternative forces might impact stability (areal spread, as insightfully discussed by the author in the discussion), it is critical to know whether the results are robust against this assumption. A straightforward way out is to perform a sensitivity analysis on trees that do not assume Transeurasian. (Areal spread for example would predict unequal rate variation across families: less when exposed to more spread, more when exposed to less spread.)

2. I am missing any model comparison, or at least model criticism. A standard approach is to compare the AIC across models (easy to compute for HMM models), with different numbers of rate categories; at least we need to know how a model with no rate variation compares to a model with rate variation. This is especially important to assess rate variation across families (see Issue #1)

3. Just to be sure I understand the "pseudo-posterior" trees: were the monophyletic constrains all and only those who are then annotated with an MCC post. prob. = 1? If so, please say so; if not, please explain in the text where the constraints were applied and what else might have led to post. prob. = 1. Also, perhaps I missed it, but where was the uncertainty used and reported for rates and D? Please make this explicit.

4. Since the trees are not timed, it is not appropriate to talk about "stable" in an absolute sense, only in the sense of "feature f_i is more/less stable than f_j ". While it looks like the author is fully aware of this, care should be taken to remove any remaining ambiguity here (e.g. p 7, opening paragraph of discussion).

5. The D metric assumes an expected signal based on a threshold model. Since such models are not standard in linguistics, this needs to be justified more explicitly, especially since the underlying assumption about a latent liability is not self-evident for language data. Also the discussion section should make clear to what extent the results depend on the threshold model assumptions; some readers might object to assumptions. Could the assumption be relaxed?

6. The correlations in Fig 4 need more attention. At least we need a separate analysis for log rates near 0. For such rates one would predict more values of D below 0 (or at least below .5) than above 0. This is hard to judge from the plot and already a quick histogram or density plot would help; better yet a formal analysis (e.g. through a hurdle or zero-inflation model)

7. For easy replicability, please turn all absolute paths to relative paths in the scripts and provide a table of contents in the readme, with a brief explanation what each script does and what each directory contains.

8. It was impossible for me to keep track of which legend belongs to which figure in the Supp. Materials; the tables appeared as latex code printed to PDF. Unreadable. Also, it would help if the captions explained briefly what the figures/tables are supposed to show and what went into them (ideally with a pointer to the code that generated the figure/table).

===PREPARING YOUR MANUSCRIPT===

one version should clearly identify all the changes that have been made (for instance, in coloured highlight, in bold text, or tracked changes);

===PREPARING YOUR REVISION IN SCHOLARONE===

-- If you are requesting an article processing charge waiver, you must select the relevant waiver option (if requesting a discretionary waiver, the form should have been uploaded, see 'File upload' above).

-- If you have uploaded any electronic supplementary (ESM) files, please ensure you follow the guidance at <https://royalsociety.org/journals/authors/author-guidelines/#supplementary-material> to include a suitable title and informative caption. An example of appropriate titling and captioning may be found at https://figshare.com/articles/Table_S2_from_Is_there_a_trade-off_between_peak_performance_and_performance_breadth_across_temperatures_for_aerobic_scope_in_teleost_fishes_/3843624.

Author's Response to Decision Letter for (RSOS-211252.R1)

See Appendix B.

Decision letter (RSOS-211252.R2)

Dear Mrs Hübler,

I am pleased to inform you that your manuscript entitled "Phylogenetic signal and rate of evolutionary change in language structures" is now accepted for publication in Royal Society Open Science.

Please remember to make any data sets or code libraries 'live' prior to publication, and update any links as needed when you receive a proof to check - for instance, from a private 'for review'

URL to a publicly accessible 'for publication' URL. It is good practice to also add data sets, code and other digital materials to your reference list.

on behalf of Kevin Padian (Subject Editor)
openscience@royalsociety.org

Appendix A

Re: Royal Society Open Science - Decision on Manuscript ID RSOS-211252

Dear Professor Padian,

I thank the reviewers for their constructive comments and suggestions, and I have implemented the majority of their suggestions as described below. In what follows, my responses are in black text, while the reviewers' comments are in blue text.

Editor Comments:

Thanks for your submission. As you see the reviews are a bit mixed but generally positive. Both reviewers have methodological questions and some concerns about interpretations. Please address these in your revision. Best wishes.

Reviewer: 1

Comments to the Author(s)

The paper presents a statistical investigation of the diachronic stability and phylogenetic signal of a range of typological features from the GramBank database, evaluated against a sizeable collection of Northern Eurasian languages.

The study is executed in a professional manner. The results are well-justified and relevant. The paper devotes adequate attention to the linguistic interpretation of the findings. For these reasons, I recommend publication.

There are a few minor comments that I would like to bring to the author's attention:

- The numerical values of the rates of change are hard to interpret. I assume that the unit here is just the average rate, but I am not sure. Anyway, interpretation of the results would be helped considerably if these rates would be set in relation to real time. This is admittedly difficult to achieve, but giving, e.g., the age of Proto-Turkic in average numbers of mutations would be very useful to relate these numbers to the real world.

The rates reported here are instantaneous rates so one can interpret them as “this is the rate of change at a single unit of branch length”. The branches are not meaningful in terms of time but one can interpret them as relatively bigger or smaller. Unfortunately, there is not enough evidence to date the full tree used here (and I am reluctant to do so especially given the controversial nature of ‘Transeurasian’). Further, using the dataset here to date the tree risks circularity by using the data to infer topology and branch lengths and then using these data on the tree to get estimates of rates and stability.

I added the following paragraph to the “Discussion”:

“The combination of reconstructed states and the rate of change of particular features can allow further research to contextualise the rates in time, if there is enough information on the age of the proto-language. For example, the age of Proto-Turkic was estimated to be around 2100 years before present (Savelyev 2020). In the first step, one extracts the features reconstructed as “present” in Proto-Turkic (95% probability of being “present” or higher). In the next step, one calculates the distance between Proto-Turkic and its children languages on these traits. This procedure provides us with a number of differences between Proto-Turkic and each child language.”

- When explaining the D statistic, it is said that a value of 0 corresponds to evolution "according to Brownian motion" (page 6). This is confusing because it applies to binary features, while Brownian motion is a dynamics of continuous features. Please clarify.

I have reworded this to now read: "D is the sum of sister clade differences across the tree and its values normally fall in the range between 0 and 1. In a trait that is strongly phylogenetically structured, sister languages will share the same value (and have no difference). If the trait is not phylogenetically patterned, then sister languages will have different values: the D value will be high and the phylogenetic signal will be low."

- page 17, lines 40/41: It is said that "[t]he 'all rates different' model obviously allows the evolutionary rate to vary in different parts of the tree". I believe the ARD model allows gain rates and loss rates to be different, but both rates are still constant across the tree. Please check.

Thank you for this correction, I deleted the statement about ARD allowing the rate to vary in different parts of the tree. The relevant part of the paragraph now reads as follows: "It is most probably due to the interaction effect between the evolutionary rate and the tip values described above: the ancestral state reconstruction is informed about the evolutionary rate of this feature, which is relatively high, and the chosen model is ARD, therefore the impact of the tips values is moderate. We get thus a rather high uncertainty in the reconstructed node value for all families."

Reviewer: 2

Comments to the Author(s)

This paper makes contributions to our understanding of the relationship between phylogenetic signal and transition rates for structural linguistic features and the stability of various parts of the grammar in 5 language families, and presents an approach for investigating the stability of structural features. Importantly, it reports a correlation between phylogenetic signal and rate of change in structural language features.

As currently written, this paper would seem to be of most interest to linguists interested in theories of relatedness among the five study families (e.g. Transeurasian or Altaic studies). There is promise here for this to be framed more generally as a blueprint for investigating stability of typological characteristics in the world's languages. However, the current manuscript foregrounds comparisons of the 5 families in the study and the big, vague, still-open question of whether typological features are stable over time and places less emphasis on the patterns of stability across areas of grammar that its approach has the ability to probe. The paper offers some useful information about similarities between families of northern Asia and a bit of evidence that tends to support prior research on what grammatical features might be relatively stable.

While this paper is a step in the direction of assessing how structural features may be used in inferring language phylogenies, as discussed in the conclusions (p. 16-17), it might be of more interest to scholars working outside of the study region/language families if the main text focused more clearly on differences in stability across areas of the grammar and made a better case for the results of this study being a) indicative of genealogical rather than areal signals, and b) generalizable to the world's languages.

Thank you for the suggestion to make the paper more interesting to scholars outside the Transeurasian/Altaic studies. While I would like to keep it relevant for them too, I acknowledge that more focus on the differences in stability across areas of grammar will make the paper more relevant for a broader linguists' audience. I implemented the following steps to highlight this paper orientation:

- 1) I introduce the approach (splitting the grammar into functional categories, levels and parts of speech) in the section “Materials” instead of the section “Discussion”
- 2) I report on the differences between the categories in the section “Results”
- 3) I revised and complemented the discussion of the implications thereof in the section “Discussion”
- 4) I added results on ancestral state reconstruction across different grammatical categories (Fig. 9)

Regarding the extent to which this study’s insights are generalizable outside of the 5 language families it focuses on, the author reflects on p. 12-13 about the problem of biases introduced by studies focused on particular language areas or families, especially w.r.t. the source of some of the linguistic data used in this paper. This raises, but does not directly address, the bigger question of whether the results of this study are truly generalizable or whether they too reflect regional patterns that may not adequately represent general processes of language evolution. What does the stability of features in these five families tell us about stability in languages of the world?

The results of the current study go in line with several known studies on stability of language structures, specifically with those on Austronesian and Indo-European (see Section “Discussion”). I present results for already 5 families, instead of only one, Austronesian, in Greenhill 2017, so the paper makes several steps forward towards a set of cross-linguistically features, if these are ever to be determined (see the discussion of why this is problematic for basic vocabulary too). In the discussion, I note that verb-final word order is reconstructed as very stable in these 5 language families, whereas verb-medial word order is unstable. This conclusion will most probably be different for different language families and there is no reconciliation to it, apart from collapsing the three features on word order (verb-initial, verb-medial, verb-final) into one feature and presenting it as stable (e.g. “Whichever position the verb takes in in the clause, this position is stable”).

I added several paragraphs in the discussion to address the generalisability of the results to other language families:

“Since we see support of the presented results in previous studies, which did not focus on the same region, and since this study covers 5 language families (albeit with hypothesised genealogical relationships or at least forming a sprachbund), we can say that stability patterns of different areas of grammar might have a cross-linguistic component. Nevertheless, we can only draw conclusions on the stability of particular features for one unit at a time, be it a language family or an area, because we can only measure features that are present in the given family or area. We cannot fully compare these results to those for Indo-European or Austronesian, because there are typical Austronesian, Indo-European and “Transeurasian” features.”

“One could think of stability not as a universal phenomenon, but as a trend that depends on the area and genealogical affiliation: one could expect language families with similar typological profiles (e.g. more synthetic or more analytic) to show similar stability patterns in their grammars, e.g. that morphology will tend to be stable if language makes use of extensive morphological marking. If the language rather uses free marking more often than bound marking, then one might hypothesise that this marking will be more stable due to its higher frequency. The current study is thus a step towards a list of universally stable structural features (if it were ever to be determined), but not the final destination in the construction thereof: it provides ground for testing new hypotheses for stability patterns in other language families.”

Some of the links between ostensibly “Transeurasian” languages have been hypothesized to reflect language contact, rather than mutual inheritance from a common ancestor. This paper’s introduction (p. 2, lines 47-48) proposes that testing the stability of features in a set of families whose

relationships may be shaped by both contact and genealogical inheritance offers an advantage, but it's not made clear what that advantage is.

Contact is little discussed in the remainder of the paper but is highlighted again in the final sentence of the conclusions. That passage (p. 17, lines 12-14) does not explain how the possibility of contact-based influence between languages in the sample confers an advantage in this research, but rather it casts some doubt on whether the findings actually demonstrate phylogenetic stability or whether they might instead tell us something about areal relationships. It would seem that focusing on a set of families thought to have been influenced by language contact might actually be a disadvantage for demonstrating phylogenetic stability of features. I don't think any historical linguist who is skeptical about the use of structural/typological features in inferring phylogenetic relationships would be reassured by the Conclusion section of this manuscript, and I believe that readers interested in language or cultural evolution outside of Transeurasian would like to see more discussion of possible areal influences throughout the interpretation of results (e.g. in the discussion of reconstruction of individual features on p. 15).

The advantage of taking a sample with a high level of contact is the following: phylogenetic signal takes into account whether or not related languages have the same feature value. The spread of the features does not necessarily occur between two closely related sister languages, but which are not spoken in nearby villages, but it does often happen between a prestigious language, say Russian, and a minority language, e.g. a Turkic language. If, despite bilingualism of the speakers in Russian and a Turkic language, that Turkic language still has the same value as its closest Turkic relative, we can say that the structural feature is stable. What we cannot account for with this method is the situation when two languages are both very closely related and stay in intense contact, but given the vast distances between speaker communities in the region, this scenario is rather rare. I added the following paragraph to the section "Introduction": "If, despite the high levels of contact between the languages in the sample and a high potential for feature transfer, we can show that some structural features have a phylogenetic signal, then it would indicate that structural features convey a historical signal that is due to genealogical relationships rather than language contact.

This reasoning also explains why the results of the study are only minimally affected by areality. I mention some cases in the section "Discussion":

"Word order in noun phrases appears to be identical in the whole area: all modifiers appear before the noun, and most of the languages in question have been in contact either with each other, or a neighboring language, often Mandarin Chinese or Russian. It is therefore difficult to conclude that constituent order is per se a genealogically stable feature, if language contact cannot contribute to variation. As for clausal word order, it has some variation in the area: all Transeurasian languages have OV order, whereas influential neighbouring languages have VO order. In some Transeurasian languages, also VO order is possible, due to borrowing, but most of the languages could resist the influence of dominant neighbouring languages and retained OV order."

"There is no gender in the languages in question, therefore we can only say that gender is difficult to gain, despite neighbouring languages (e.g. Russian) having gender."

I discuss how this is reflected in the numbers for the evolutionary rate and D in the following paragraph:

"In her 1992 book, Nichols mentioned word order as being very genetically unstable. This finding is only partially supported by my results: the category "word order" as a functional category evolves at a slow rate ($q_{01} = -0.52$ and $q_{10} = -1.17$) and has a high phylogenetic signal ($D = -0.05$), but the stability varies for particular orders of constituents (OV is more stable than VO). Nichols

(1993: 353) classifies clause word order as areally stable, and this is definitely true for the current language sample: the word order is verb-final in all languages in the sample ($D = -4.95$, $q01 = -10$, $q10 = -10$), with 4 languages (Gagauz, Khalaj, Beryozovka Even and Moghol) also allowing verb-medial word order due to borrowing ($D = 1.07$, $q01 = 0.06$, $q10 = 1.21$). It is difficult to conclude that verb-final word order is more stable than verb-medial word order: in a language sample, where 60 out of 60 languages have verb-final word order, this feature has not changed, but this might well be the case for verb-medial word order in an area or a language family, where this particular order dominates.”

Finally, I believe that presenting the findings regarding stability profiles of different sets of structural features more prominently would generate more interest across a broad audience that may not have much familiarity with Transeurasian languages. The feature categories are not introduced until the discussion section, and even then the justification for splitting the feature set into these particular functional categories and levels of grammar used is not made clear. Until this is discussed, reader interpretation of results is hampered by a lack of clarity about the nature of the data being analyzed.

I have added a description of feature categorisation in the subsection “Materials” as requested.

Some other smaller points follow:

There is a long discussion on p.6 (lines 32-59) of phylogenetic signal D values that might be better presented in tabular format than prose.

Moved this discussion into Table 1.

The discussion of D value ranges on p. 5, lines 56-58 repeats information already conveyed on p. 4.

Removed the ranges in the discussion.

On p. 15, the potential for missing data to impact results is discussed. It would be useful to see an earlier mention of how much data is missing in this dataset. This is also important for a reader's understanding of the results.

I have added the following statement to the “Materials” section, and included plots with missing data and D/rates in the electronic supplementary materials (ESM, Fig. 2):

“Out of the 171 features, more than a half of the languages could be coded for 95% of features (162 features), around two third of the languages could be coded for more than 78% of features (134 features) (see ESM, Fig. 1 for the relationship between the amount of “present”, D and rates and Fig. 2 for the relationship between missing data, D and rates).”

Figure 2 labels internal nodes only with probabilities. I recommend also including family labels for the nodes that represent the proto-languages of the 5 families (as in Fig. 10).

Included language family names in Fig. 2.

In several plots you show information about transition rates for feature gain and loss. These are labeled with “0->1”/“0 to 1” and “1->0”/“1 to 0” respectively, but it might be useful for a general audience to also include “(gain)” and “(loss)” in those figure labels (Fig. 3, 4, 6, 7, 8).

Included “gain” and “loss” in figures 3-7 and in all tables.

There are a number of typos throughout that need to be found and fixed. For example, p. 9, line 31 repeats the word “the” and the caption for Figure 5 spells “present” as “prezent”. I recommend another round of editing for typos and minor grammatical matters.

I have fixed these specific cases and the paper has now received a thorough proofreading by several colleagues.

Appendix B

Re: Royal Society Open Science - Decision on Manuscript ID RSOS-211252.R1

Dear Professor Padian,

I thank the reviewer for these remarks. I addressed them as described below and adjusted the text and figures where it appeared possible. In what follows, my responses are in black text, while the reviewers' comments are in blue text.

Editor Comments:

Thanks for your revision, which seems to have satisfied most of the reviewers' concerns. There seem to be some remaining points to clarify, and if you require more time for this please let our Editorial Office know. Best wishes.

Reviewer comments to Author:

Reviewer: 3

Comments to the Author(s)

This is a great paper and it will move the field ahead. There are some remaining issues though that need to be taken care of before the results can be considered sufficiently robust and clear:

1. The methods presuppose the existence of the phylogeny that they are used on, i.e. that all languages trace back to a common ancestor. Since this is heavily disputed in the case of Transeurasian and because alternative forces might impact stability (areal spread, as insightfully discussed by the author in the discussion), it is critical to know whether the results are robust against this assumption. A straightforward way out is to perform a sensitivity analysis on trees that do not assume Transeurasian. (Areal spread for example would predict unequal rate variation across families: less when exposed to more spread, more when exposed to less spread.)

The trees integrate over all possible configurations of “Transeurasian”, i.e. I am not making any claims about the validity of TE or of any deep relationships within these languages. There is no assumption for or against TE in my analysis. The high probability value (1.0) on the highest node is an artefact of the fact that no other languages are included in the sample. Note that none of the internal groupings have high probabilities (no Mongolo-Turkic or Tunguso-Mongolo-Turkic or similar).

I modified the caption of Figure 2 to read as follows: “high probability on the node (1) indicates the monophyletic constraint on the node according to the Glottolog [v.4.2] classification, except for the root probability value (1), which is an artefact of the fact that no languages, apart from Transeurasian, were included in the sample (i.e. there is no outgroup).”

A follow-up project would be to use these data to test TE, but it would require a wider collection of data from other languages outside TE (e.g. Uralic, or Austro-Asiatic) and is outside the scope of this paper.

2. I am missing any model comparison, or at least model criticism. A standard approach is to compare the AIC across models (easy to compute for HMM models), with different numbers of rate categories; at least we need to know how a model with no rate variation compares to a model with rate variation. This is especially important to assess rate variation across families (see Issue #1)

I added the results of the “equal rates” model to the supplementary materials. I decided not to do a formal model test because this would require one model test per feature per pair (1000 trees * 171 features), which is problematic in terms of massive multiple testing. I am especially interested in the directionality of change and have strong a priori reasons to believe the rates of gain and loss are different in linguistic data. The differences in borrowability and persistence of items lie at the core of this paper and we would lose an immense amount of information if we were to consider an equal rate. My results show that features covering the phonological shape and word domains are lost at a considerably slower rate than they are gained (-1.02 vs. -0.36 and -1.22 vs. -0.92). We see even more striking differences between the rate of loss and gain in functional categories: here deixis is lost at a rate of -0.52 and gained at a rate of 0.27, TAME+ features are lost at a rate of -0.22 and gained at a rate of 0.02. Among parts of speech, the differences between rates are highest for adjective (-0.35 and 0.08), article (-0.45 and 0.92), demonstrative (-0.3 and 0.61), and particle (-0.58 and 0.62).

I extended the motivation for the model choice in the “Methods” section to read as follows: “As there are no strong grounds to assume that linguistic features are gained and lost at the same rate and as the differences between these rates will be particularly interesting for explaining linguistic diversity in future research, I chose the model ARD.”

3. Just to be sure I understand the "pseudo-posterior" trees: were the monophyletic constrains all and only those who are then annotated with an MCC post. prob. = 1? If so, please say so; if not, please explain in the text where the constraints were applied and what else might have led to post. prob. = 1.

In the methodology section, there is the following explanation of the probabilities on the MCC tree: “This procedure provided a posterior probability distribution of the trees with each node in the Glottolog classification having a posterior probability of 1.0”. I changed part of the caption of the Figure 2 to read as follows: “The maximum clade credibility tree: low probability on the node indicates either an agnostic view on the order of the splits or that a branch split into more than 2 further branches (i.e. a non-binary structure of the tree); high probability on the node (1) indicates the monophyletic constraint on the node according to the Glottolog [v.4.2] classification, except for the root probability value (1), which is an artefact of the fact that no languages, apart from Transeurasian, were included in the sample (i.e. there is no outgroup).”

Also, perhaps I missed it, but where was the uncertainty used and reported for rates and D? Please make this explicit.

I added standard deviations to the table in the supplementary materials and split it into three tables: Table S2 for D, Table S3 for rates, Table S4 for states. A table with raw values, before the log-transformation of the rate, can be found on github, in the folder “results” (SI_summary_table).

4. Since the trees are not timed, it is not appropriate to talk about "stable" in an absolute sense, only in the sense of "feature f_i is more/less stable than f_j ". While it looks like the author is fully aware of this, care should be taken to remove any remaining ambiguity here (e.g. p 7, opening paragraph of discussion).

I changed the opening paragraph of the Discussion to read as follows: "In the terms used in this study, a feature is stable if it evolves at a relatively slow rate and has a high phylogenetic signal. The results have shown that 66% of the features in the data set have a D value below 0.5 and evolutionary rate below 0 (after the log₁₀ transformation). Therefore, more than half of the features can be called relatively stable."

5. The D metric assumes an expected signal based on a threshold model. Since such models are not standard in linguistics, this needs to be justified more explicitly, especially since the underlying assumption about a latent liability is not self-evident for language data. Also the discussion section should make clear to what extent the results depend on the threshold model assumptions; some readers might object to assumptions. Could the assumption be relaxed?

This is an implementation detail of the particular metric that enables it to be robustly calculated. The assumption cannot be relaxed, but other metrics could be used, however they have much lower power and are unlikely to provide qualitatively different results.

6. The correlations in Fig 4 need more attention. At least we need a separate analysis for log rates near 0. For such rates one would predict more values of D below 0 (or at least below .5) than above 0. This is hard to judge from the plot and already a quick histogram or density plot would help; better yet a formal analysis (e.g. through a hurdle or zero-inflation model)

I revised the figure and removed the regression line from it. I removed the values with rates equal to 0 (or -10 after the log₁₀ transformation) from the figure and added two histograms in the supplementary materials with the distribution of D values for feature gain and loss rates (Figure 7). I added the following explanation in the text of the Results section: "There are 58 (gain) and 34 (loss) features that have rate equal to 0 prior to log₁₀ transformation. These features are thus almost never lost or gained: they are either present in all languages (e.g. 56 out of 56 languages with enough information on the feature or 60 out of 60 languages, i.e. the whole sample) or absent in all but one or two languages (e.g. "present" in 1 out of 60 or 2 out of 24 languages). For the distribution of the D values across these features, see Figure 7 in ESM."

I deleted the following text from the Figure 4 caption: "The accumulation of the values at the far-left end is due to the infinite values introduced through the log₁₀-transformation. This issue concerns low-information features, where most of the languages have the same value: either almost all "present" or almost all "absent". They are almost never lost and almost never gained and their rate of change equals zero." and added the following explanation: "Features with the rates equal to zero prior to the log₁₀ transformation (58 for feature gain and 34 for feature loss) are not included in the plot: these features are either present in all languages or absent in all but one or two languages."

7. For easy replicability, please turn all absolute paths to relative paths in the scripts and provide a table of contents in the readme, with a brief explanation what each script does and what each directory contains.

I changed the paths from absolute to relative, added a table of contents in the readme on GitHub, sorted the files into folders (e.g. scripts, plots, data, results, etc.) and adjusted the scripts to retrieve data from these folders and to save the output in correct folders.

8. It was impossible for me to keep track of which legend belongs to which figure in the Supp. Materials; the tables appeared as latex code printed to PDF. Unreadable. Also, it would help if the captions explained briefly what the figures/tables are supposed to show and what went into them (ideally with a pointer to the code that generated the figure/table).

I was aware of the fact that tables were included as latex code in the compiled pdf document, but, unfortunately, multiple uploading attempts to the submission system with different constellations did not remedy it. I have both a compiled pdf file with all supplementary materials and, separately, all the raw tables, as I was requested to submit them upon revision. I tried to upload them all during the submission, but only the raw tables were inserted in the final pdf document. I hope this will not happen during the next submission.

I complemented the captions of the tables in the supplementary materials with more details on the variables in them and what values in particular ranges mean.

I created an R-Markdown file from the script `explore_results.R`, which recreates all tables, statistics and almost all plots (apart from the heatmap, which is included in a separate short script).